# Derailed protein turnover in the aging mammalian brain

Nalini R Rao[1,2], Arun Upadhyay [1,2] & Jeffrey N Savas [1✉]

## Abstract

Efficient protein turnover is essential for cellular homeostasis and organ function. Loss of proteostasis is a hallmark of aging culminating in severe dysfunction of protein turnover. To investigate protein turnover dynamics as a function of age, we performed continuous in vivo metabolic stable isotope labeling in mice along the aging continuum. First, we discovered that the brain proteome uniquely undergoes dynamic turnover fluctuations during aging compared to heart and liver tissue. Second, trends in protein turnover in the brain proteome during aging showed sex-specific differences that were tightly tied to cellular compartments. Next, parallel analyses of the insoluble proteome revealed that several cellular compartments experience hampered turnover, in part due to misfolding. Finally, we found that age-associated fluctuations in proteasome activity were associated with the turnover of core proteolytic subunits, which was recapitulated by pharmacological suppression of proteasome activity. Taken together, our study provides a proteome-wide atlas of protein turnover across the aging continuum and reveals a link between the turnover of individual proteasome subunits and the age-associated decline in proteasome activity.

**Keywords** Aging; proteasome; protein turnover; quantitative proteomics; stable isotope labeling
**Subject Categories** Post-translational Modifications & Proteolysis; Proteomics; Translation & Protein Quality

## Introduction

Aging is a multifaceted and nonlinear process in which cell and organ functionality declines. One hallmark of aging is impaired protein homeostasis, or proteostasis, which is a delicate balance between protein synthesis, folding, and degradation (Labbadia and Morimoto 2015; Vilchez et al, 2014). A critical component of proteostasis is protein turnover, which is achieved through waves of synthesis and degradation (Fornasiero and Savas 2023). Deficits in protein turnover lead to protein misfolding, accumulation, and impairment of cellular function.

Perturbations in the proteostasis machinery play an important role in modulating aging-associated pathways (Balch et al, 2008; Hipp et al, 2019; Jayaraj et al, 2020; Klaips et al, 2018; Labbadia and Morimoto 2015; Morimoto and Cuervo 2014; Vilchez et al, 2014). Studies in invertebrates using genetic or pharmacological tools to promote proteasome activity or clearance of deposits have found that reducing proteotoxic stress can extend lifespan (Baird et al, 2014; Balch et al, 2008; Klaips et al, 2018; Leeman et al, 2018; Vilchez et al, 2014). Similar interventions in mammals aimed at slowing aging by enhancing proteasomal degradation have begun to show similar results, further implicating protein turnover in aging (Uno and Nishida 2016; Upadhyay 2021). However, we lack a deep understanding of the molecular pathways underlying aging, as the majority of proteomic studies investigating mammalian aging are based on comparisons between young and old animals (Angelidis et al, 2019; Kelmer Sacramento et al, 2020; Kluever et al, 2022; Yousefzadeh et al, 2021). While these studies capture important snapshots of age-associated changes in relative protein abundance, they do not capture aging in a continuous manner or examine the dynamics of protein turnover.

Maintaining balanced protein turnover is particularly important in post-mitotic cells because these cells cannot dilute damaged proteins by division. Previous research investigating protein turnover in rodents using [15]N-containing amino acids has identified unexpected pools of proteins with particularly long lifetimes in tissues containing post-mitotic cells, such as heart, eye lens, and brain (Bomba-Warczak et al, 2021; Savas et al, 2012; Toyama et al, 2013). Previous studies of protein lifetimes in the brain using short-term labeling with lysine chow have shown that aged animals have increased protein lifetimes compared to young animals (Kluever et al, 2022). However, since the brain contains abundant post-mitotic cells, it is particularly important to examine the dynamics of protein turnover during aging in the long-lived proteome. Furthermore, our previous studies focusing on mammalian models of Alzheimer's disease (AD) have shown that impaired protein turnover at presynaptic sites is one of the earliest detectable pathologies (Hark et al, 2021). In addition to AD, deficits in protein turnover have been documented in other neurodegenerative diseases, as evidenced by the presence of proteotoxic aggregates and protein deposits in humans and mammalian models (Auluck et al, 2002; Balch et al, 2008; Hark et al, 2021; Kurtishi et al, 2019; Savas et al, 2017). Importantly, age is the largest known risk factor for most neurodegenerative diseases. While it is clear that impaired protein turnover is closely linked to age-related diseases, less is

[1]Department of Neurology, Northwestern University Feinberg School of Medicine, Chicago, IL 60611, USA. [2]These authors contributed equally: Nalini R. Rao, Arun Upadhyay.
✉E-mail: jeffrey.savas@northwestern.edu

known about the proteome-wide changes in turnover that occur across the aging continuum. Developing an understanding of protein turnover changes combined with robust temporal resolution will further our understanding of healthy aging. In addition, studying protein turnover during aging may elucidate subcellular compartments with greater susceptibility to age-related disease impairments and functional losses.

In this study, we established a continuous pulse-step metabolic labeling paradigm in mice and used quantitative proteomic analyses to uncover age-related fluctuations in protein turnover. We discovered the cortical proteome exhibits sex-specific, highly dynamic turnover during aging that is associated with distinct cellular compartments. We then investigated whether protein groups with different turnover trends were due to protein misfolding. Analysis of insoluble proteins revealed age-dependent specific protein turnover fluctuations. To further probe whether the hampered turnover of the insoluble protein pool was a result of dysfunction of the ubiquitin proteasome system (UPS), we biochemically isolated proteasomes. This revealed fluctuations in proteasome activity that correlated with the turnover of the catalytically active proteasome subunits. Taken together, our findings uncover proteome-wide changes in turnover during aging and reveal that protein turnover in the cortex is a non-linear process that undergoes sex-specific and cellular compartmental fluctuations that may be caused by reduced proteasome fidelity.

## Results

### The brain proteome turns over slower than the heart and liver proteomes over the aging continuum

To study protein turnover during organismal aging, we designed a continuous in vivo metabolic labeling paradigm in mice. We metabolically labeled five groups of male and female C57BL/J6 mice with $^{15}N$ chow in three-month increments beginning at 9 months of age (early adulthood) through 21 months of age (aged mice). This allowed us to capture proteome changes in a continuous manner with the stepwise $^{15}N$ labeling periods from 9–12, 12–15, 15–18, 18–21, and 21–24 months of age (Fig. 1A). At the end of each labeling period, animals were euthanized, harvested, and subjected to in-depth proteomic analysis. For this analysis, we focused on two regions known to harbor post-mitotic cell types (cortex and heart) and one tissue with highly proliferative cells (liver) from female cohorts. Proteins with relatively fast turnover rates during the three-month labeling periods will be enriched with $^{15}N$ atoms and represent the newly synthesized protein pools. Proteins with slower turnover rates will persist throughout the labeling periods and remain composed of $^{14}N$ atoms, thus represent old proteins ($\geq 3$ months). This paradigm allows us to determine changes in relative protein turnover with robust temporal resolution across the aging continuum.

Cortex, heart, and liver tissue homogenates were analyzed by liquid chromatography-tandem mass spectrometry (MS)-based proteomic analysis to identify and quantify the amount of fully $^{15}N$ (new; $\leq 3$ months) and $^{14}N$ (old; $\geq 3$ months) labeled proteins. Consistent with our previous studies using a similar labeling period, cortical, heart, and liver proteins were predominately $^{15}N$ labeled (Bomba-Warczak et al, 2021; Hark et al, 2021; Savas et al, 2012). No

significant differences in the number of newly synthesized ($^{15}N$; $\leq$ 3 months) protein identifications were observed between the age groups and only heart and liver tissues showed tissue differences (Fig. 1B). In contrast, the identification of $^{14}N$ (old; $\geq 3$ months) proteins was substantially higher in cortical extracts compared to heart and liver (Fig. 1C). In addition, consistent with our previous findings, heart tissue also contained significantly more $^{14}N$ (old; $\geq 3$ months) proteins compared to liver (Bomba-Warczak et al, 2021). $^{14}N$ fractional abundance (FA)—a metric of relative $^{14}N$ protein turnover - was then calculated based on reconstructed MS1 chromatograms to quantify the amount of $^{14}N$ (old; $\geq 3$ months) proteins remaining relative to the total pool of proteins ($^{14}N$ and $^{15}N$). Accordingly, a higher $^{14}N$ FA value indicates a greater proportion of old proteins relative to the total protein pool. PCA analysis revealed reproducible $^{14}N$ FA values among the biological replicates for each of the five age groups (Fig. EV1A,B). $^{14}N$ FA calculation of the identified $^{14}N$ proteins revealed that the cortical proteome has the highest overall content of $^{14}N$ proteins (old; $\geq 3$ months), with an average $^{14}N$ FA of $8 \pm 1.18\%$ across all aging time points, closely followed by heart tissue with $7 \pm 0.19\%$. In contrast, liver consistently showed the lowest average $^{14}N$ FA of $<1 \pm 1.09\%$ across aging groups, indicating that less than 1% of the proteome persists in liver throughout all the labeling steps (Fig. 1D). Since both cortex and heart had a substantial pool of $^{14}N$ proteins, we next quantified the percentage of $^{15}N$ incorporation for each identified protein in each age group. We plotted this distribution to begin to understand if there are proteome-wide changes in $^{15}N$ incorporation rates during aging (Fig. EV1C,D). Interestingly, although both cortex and heart contain a substantial pool of $^{14}N$ proteins, only the cortical proteome showed significant differences in $^{15}N$ incorporation during aging. To directly compare this phenomenon between tissue types and age groups, we selected a representative protein, malate dehydrogenase 2 (Mdh2), present in cortex, heart and liver. We then extracted the raw $^{14}N$ and $^{15}N$ peptide MS1 spectra for Mdh2 and found that this protein showed a fluctuating $^{14}N$ abundance during aging in cortical extracts (Fig. 1E). In contrast, Mdh2 protein showed a stable $^{14}N$ abundance in heart tissue, while liver showed no identifiable $^{14}N$ peaks.

### Sex-specific and cellular compartmental differences in cortical protein turnover across the aging continuum

The data so far indicate that proteome-wide turnover in the cortex, but not in the heart or liver, varies across the aging continuum. Age-related changes in the brain proteome have been previously described and many neurodegenerative diseases differentially affect males and females (de Jong et al, 2023; Lemaitre et al, 2020; McCartney et al, 2019; Palliyaguru et al, 2021; Ward et al, 2018). Thus, we performed an extensive protein turnover analysis of cortical extracts across the five age groups and between sexes. To obtain accurate measurements of relative protein turnover using the same set of peptides, we performed MS3-based quantitative proteomic analysis of bulk cortical extracts using tandem-mass tags (TMT). We performed four 10-plex TMT-MS experiments to quantify differences during aging and between the sexes (Fig. 2A and Fig. EV2A). We quantified 602 and 421 $^{14}N$ proteins for male and female datasets respectively (Fig. EV2B), and of these $^{14}N$ proteins, 374 were common to both sexes (Fig. EV2C).

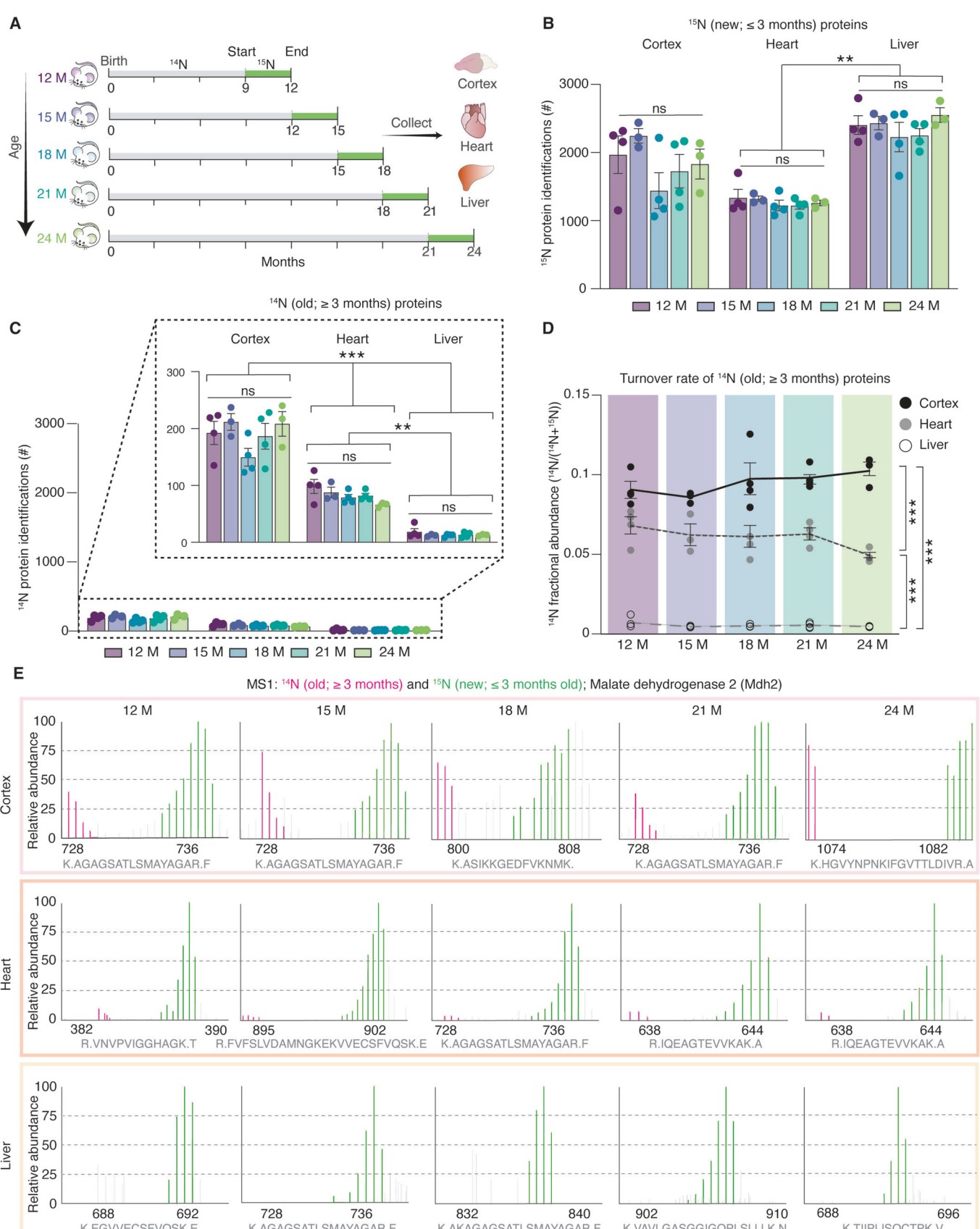

**A** Schematic showing birth, $^{14}N$ and $^{15}N$ labeling periods with Start and End for ages 12 M, 15 M, 18 M, 21 M, 24 M. Collect Cortex, Heart, Liver.

**B** $^{15}N$ (new; ≤ 3 months) proteins

**C** $^{14}N$ (old; ≥ 3 months) proteins

**D** Turnover rate of $^{14}N$ (old; ≥ 3 months) proteins

**E** MS1: $^{14}N$ (old; ≥ 3 months) and $^{15}N$ (new; ≤ 3 months old); Malate dehydrogenase 2 (Mdh2)

**Figure 1. Continuous stepwise stable isotope labeling reveals brain proteins with dynamic protein turnover across aging compared to heart and liver.**

A Whole animal metabolic ¹⁵N labeling scheme to measure protein turnover dynamics across aging in mouse brain (i.e., cortex), heart, and liver. C57BL/J6 mice were fed ¹⁵N chow from: 9–12, 12–15, 15–18, 18–21, and 21–24 months. B, C Summary of ¹⁵N and ¹⁴N protein identifications based on MS2 spectra for each age group and tissue. D Throughout aging, cortex and heart retain a significantly larger proportion of the proteome (¹⁴N FA) that is fully ¹⁴N labeled compared to heart and liver. Proteins identified as ¹⁴N or ¹⁵N peptides are included and quantified based on reconstructed MS1 chromatograms. E Annotated representative raw MS1 spectra across the indicated m/z ranges after 3 months of ¹⁵N labeling. Distinct peptides for the Mdh2 protein are shown; green indicates ¹⁵N; pink indicates ¹⁴N; gray is other. All data are mean ± SEM with *n* = 3–4 female mice. *p value < .05; **p value < 0.01; ***p value < 0.001 by Kruskal–Wallis ANOVA with Tukey's multiple comparisons test or 2-way ANOVA.

We first examined whether there were differences in the abundance of ¹⁵N (new; ≤3 months) proteins and found that there were no significant differences in either males or females during aging (Fig. 2B,C). While the abundance of newly synthesized proteins did not differ during aging in either sex, quantification of ¹⁴N FA showed significant differences in both male and female proteomes during aging (Fig. 2D,E). In males, ¹⁴N FA was highest at the 15 M time point, whereas females showed a peak at the older 21 M time point. Visualization of ¹⁴N FA for each quantified protein in males and females revealed striking trend variations (Fig. 2F,G, Dataset EV1). We used k-means clustering to elucidate distinct patterns of ¹⁴N FA and identified groups of proteins that followed similar turnover trends during aging (Fig. EV2D,E). To investigate whether proteins with similar turnover during aging were associated with shared cellular compartments and/or function, we subjected each protein cluster in both male and female datasets to gene ontology (GO:CC) enrichment analysis (Fig. 2H, Dataset EV2). Confirming our hypothesis, clusters were differentially enriched for terms such as: 'synapse', 'mitochondria', 'chromatin' and 'membrane envelope'. This finding confirms that the cortical proteome undergoes diverse and non-linear protein turnover trends during aging and shows that groups of functionally related proteins undergo similar turnover during aging. Notably, in both male and female datasets, we identified unique clusters that contained proteins with temporally distinct turnover trends but were enriched in common cellular compartment terms. For example, male cluster 3 proteins exhibited a peak in ¹⁴N FA at 15 M and were significantly enriched for the GO:CC terms: 'synapse', 'mitochondria' and 'myelin'. Similarly, female cluster 1 proteins showed robust enrichment for the same three GO:CC terms, but the slowed turnover peak for this group was shifted to a later 21 M time point.

## Protein misfolding contributes to differential trends in protein turnover in the cortex

Since we observed that distinct groups of proteins underwent variable turnover during aging, we next asked whether these differences were a consequence of protein misfolding, which prevents efficient degradation of ¹⁴N proteins. To address this question, we biochemically isolated the SDS-resistant insoluble fraction (i.e., insolubleome) from cortical extracts (Fig. EV3A). Using quantitative proteomics, we identified insoluble proteins in male and female datasets and found that 378 proteins were present in both sexes (Fig. EV3B–D). As our initial proteome-wide turnover analysis was performed on bulk cortical extracts, we investigated whether the differential turnover trends of ¹⁴N proteins identified were driven by insoluble proteins. Quantification of insoluble ¹⁴N FA in male and female datasets showed no overall

significant differences across aging (Fig. 3A,B). However, since previous studies have shown that a fraction of the proteome remains consistently misfolded and insoluble, we sought to separate age-independent from age-dependent changes in turnover (Shen et al, 2021; Vecchi et al, 2020). A protein with a high ¹⁴N FA value indicates that a large fraction of this protein pool remains uniformly long-lived during aging. We hypothesized that known long-lived proteins, such as those in the nucleosome and nuclear pore complex, would have relatively high ¹⁴N FA throughout aging. Clustering analysis yielded groups of insoluble proteins with similar rates of turnover across aging in both sexes and differentiated groups of proteins with higher and lower ¹⁴N FA across aging (Fig. 3C,D and Dataset EV3). In both sexes, we discovered distinct trends in insoluble protein turnover, represented by the blue, purple, and pink clusters. To determine whether the proteins in each cluster were overrepresented with common cellular compartments, we subjected each to GO:CC analysis. We found that the proteins in purple and pink groups in both sexes were enriched for terms, such as 'synapse' and 'postsynapse' (Fig. 3E,F). This is consistent with our previous results showing that although the timing of protein turnover fluctuations are different between the sexes, the cellular compartments affected are similar. In contrast, the blue clusters in both sexes were overrepresented for GO:CC terms such as 'nucleosome' and 'extracellular matrix' (Fig. 3E,F). As previously established, a portion of the proteome is insoluble (Toyama et al, 2013; Vecchi et al, 2020). The identification of proteins with the highest ¹⁴N FA may represent the pool that misfold in an age-independent manner. In contrast, the presence of purple and pink clusters indicates a possible age-dependent inefficiency in protein turnover.

The distinct segregation of age-dependent and independent protein pools led us to ask if there were also common intrinsic biophysical properties underlying the propensity to become insoluble. To determine this, we calculated energy minima and solubility scores for each protein in the blue cluster compared to the pink and purple clusters, as these two clusters shared a very similar pattern of GO:CC enrichment. A lower best energy and solubility value indicates a higher amyloidogenic tendency. We found that in the male datasets, proteins from the pink and purple clusters had significantly lower solubility and free energy, demonstrating that these proteins have more intrinsically amyloidogenic biophysical properties (Fig. EV3E). This phenomenon, although not significant, was comparable in females (Fig. EV3F). Since proteins from the pink and purple groups had a higher amyloidogenic propensity, we postulated that any inefficiencies in cellular turnover pathways during aging would leave them highly susceptible to misfolding. Previous studies have established that disruptions in ubiquitination or the UPS lead to protein misfolding (Elsasser and Finley 2005; Mayer 2003; Segref and Hoppe 2009). In

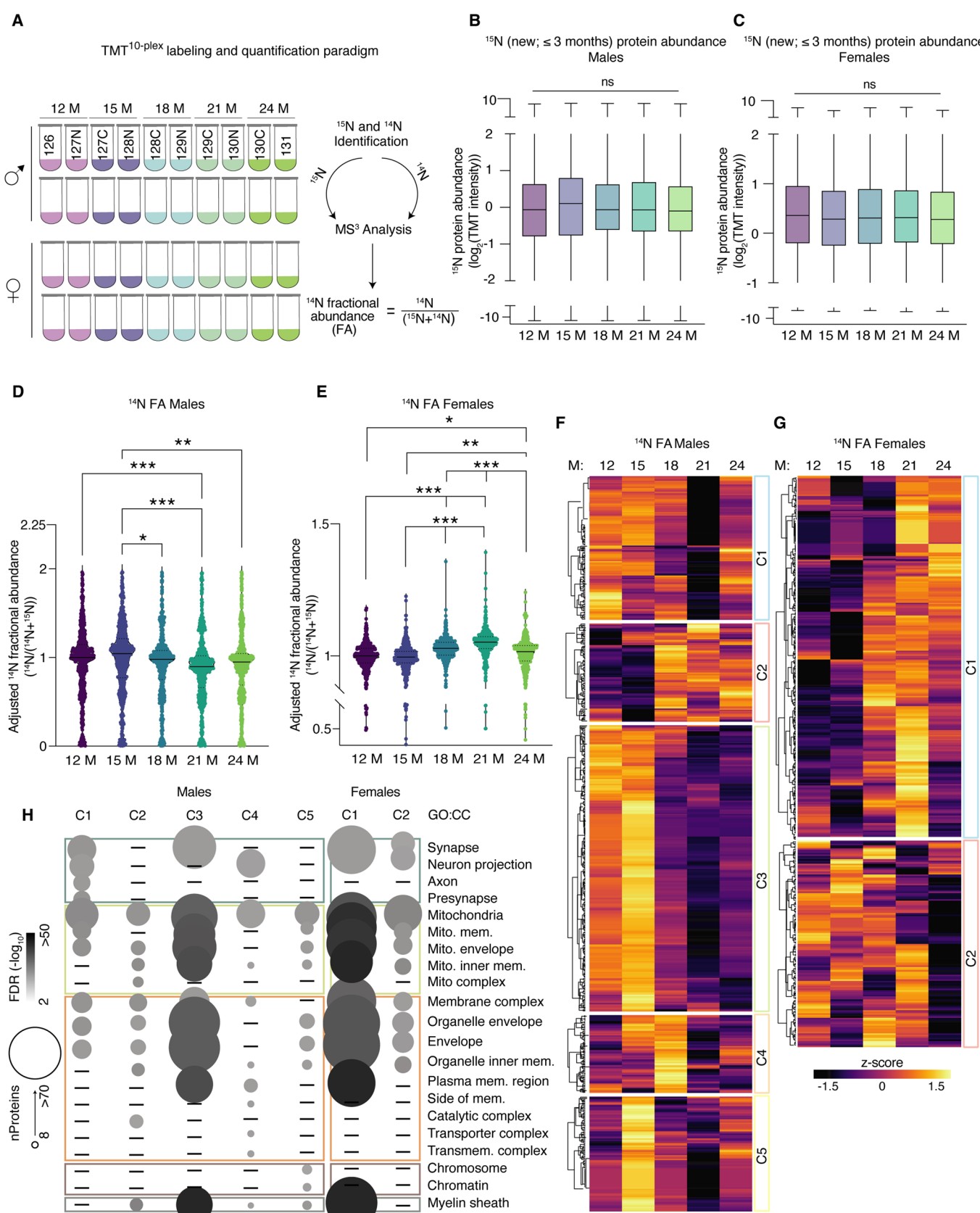

**Figure 2. Sex-specific and cellular compartment differences in cortical protein turnover across the aging continuum.**

**A** Experimental design using ten-plex tandem-mass tags (TMT) and synchronous precursor selection (SPS)-MS³-based proteomic analysis. $^{14}$N fractional abundance (FA) is calculated based on $^{14}$N and $^{15}$N reporter ion intensities and used is as a metric of relative protein turnover in male and female cortical extracts. **B, C** Global $^{15}$N protein abundance based on TMT reporter ion intensities from both male and female cohorts shows no significant differences. **D, E** Global $^{14}$N FA analysis from both male and female cohorts shows significant differences across aging. Box plot center is median with 25th to 75th percentile bounds. Whiskers extend from smallest to largest values. **F, G** Heatmap illustrating the $^{14}$N FA (z-score) and k-means clustering analysis to visualize trends during aging. Five clusters, (C1-C5), were identified across the male dataset, while two clusters, (C1 and C2), were detected in the female datasets. **H** Gene ontology enrichment analysis of each respective cluster revealed significantly enriched cell component (CC) terms. mem = membrane; mito = mitochondria. All data are mean ± SEM with $n = 4$ mice for both sexes except $n = 3$ for female 15 M and 24 M groups. $^{14}$N FA is adjusted to median of 12 M group and scaled by row (z-score) for heatmap representation. *$p$ value < 0.05; **$p$ value < 0.01; ***$p$ value < 0.001 by Kruskal–Wallis ANOVA with Tukey's multiple comparisons test. For k-means analysis, k groups were chosen based on optimal silhouette score.

support of this possibility, we identified the subset of ubiquitinated insoluble proteins (Fig. EV3G,H). Insoluble proteins harboring a diGly ubiquitin remnant (K-ε-GG) represent proteins that were marked for degradation but escaped cellular degradation pathways. Quantification of turnover for this subset of ubiquitinated insoluble proteins provides a highly reliable metric of efficiency in cellular proteostasis during aging. Remarkably, the $^{14}$N FA of ubiquitinated proteins was significantly reduced in males at the 21 M time point, and the pool of insoluble ubiquitinated proteins was heavily composed of synaptic proteins (Fig. 3G,H). This phenomenon was not driven by a decreased abundance of ubiquitinated proteins in the insoluble pool (Fig. EV3I).

## Reduced proteasome activity is associated with slowed turnover of 20 S subunits during aging

Since we identified insoluble ubiquitinated proteins with fluctuating turnover rates, we next asked whether this could be due to age-related changes in UPS activity. To answer this question, we purified proteasomes from cortical extracts and confirmed the enrichment of proteasomes by Western blot using a Psmc5 antibody (Fig. 4A,B). This strategy allowed us to quantify turnover rates for individual proteasomal subunits and potential interacting partners. First, we quantified the $^{14}$N FA of the purified proteasomal material during aging. We found a significant age-dependent linear increase in $^{14}$N FA, suggesting the age-dependent accumulation of undegraded $^{14}$N proteins (Fig. 4C). This was not due to a decrease in total proteasomal subunit expression (Fig. 4D). Therefore, we speculated that the increased protein accumulation was, at least in part, due to a decrease in proteasome enzymatic activity.

To test this possibility, we performed ubiquitin C-terminal 7-amido-4-methylcoumarin (Ub-AMC) substrate-based deubiquitinase (DUB) activity assay to first examine the specific efficacy of the 19 S regulatory complex (19 S RP). Contrary to previous reports, we found a significant relative increase in DUB activity in mice at the 21 M and 24 M time points (Fig. 4E) (Gray et al, 2003; Kelmer Sacramento et al, 2020). Interestingly, anti-ubiquitin Western blot analysis also showed the highest levels of ubiquitinated proteins in the purified material at the 21 M and 24 M time points (Fig. EV4A). Consistent with our previous observation, this indicates an age-dependent accumulation of ubiquitinated proteins despite high DUB activity. To determine whether the reduced proteasome-mediated degradation was instead related to 20 S function, we performed a 20 S proteasome-specific activity assay. We detected significantly decreased activity at the 12 M and 24 M time points compared to the 18 M time point, indicating impaired proteolytic degradation (Fig. 4F). While this phenomenon could be partly the

result of continued protein accumulation within the proteasome core, it could also be a consequence of impaired turnover of the proteolytic core subunits. We hypothesized that if the proteolytic subunits of the 20 S core are not replenished during aging, the efficiency of the 20 S core will decrease. Therefore, we quantified $^{14}$N FA of proteasomal subunits at each age (Fig. EV4B and Dataset EV4). Notably, the $^{14}$N FA for the 20 S core was significantly lower throughout aging compared to the 19 S RP (Fig. 4G). This suggests that the 20 S core is replenished more rapidly than the 19 S RP. Interestingly, when the 19 S RP and 20 S core activity is highest, 21 M and 18 M respectively, the $^{14}$N FA is lowest during aging. In support of this finding, we observed that the catalytic subunits (Psmb1, Psmb2, and Psmb5) of the 20 S core had a significantly less $^{15}$N abundance at the 24 M time point compared to the 18 M timepoint (Fig. 4H). This finding confirms that the 18 M time point represents the age with the most newly synthesized catalytic subunits and a corresponding peak in 20 S activity.

## Pharmacological suppression of 20 S proteolytic activity causes slower proteasome subunit turnover

During aging, we observed a strong correlation between proteasome activity and subunit turnover. Therefore, we next established an in vivo paradigm to directly assess whether perturbation of proteasome activity modulates proteasome subunit turnover. To inhibit proteasome activity, we used the irreversible and brain penetrant drug marizomib (salinosporamide A), which binds to 20 S core subunits Psmb1, 2, and 5 to block activity (Kelmer Sacramento et al, 2020; Manton et al, 2016). To quantify turnover after dampening proteasome activity, we used stable isotope labeling with heavy $^{13}$C6-lysine (Lys) chow. At 18 M, when we previously identified the highest peak in 20 S core activity, cohorts of male mice received an i.p. injection of 100 μg/kg marizomib or vehicle. After 24 h, all cohorts were switched from normal chow ($^{12}$C6) to a $^{13}$C6 lysine diet for 7 days prior to tissue collection for downstream analyses (Fig. 5A). We chose this time frame since previous studies have found that proteasomal subunit lifetimes are on average about 8 days (Fornasiero et al, 2018; Kluever et al, 2022). As expected with partial proteasome inhibition, Western blot analysis confirmed the accumulation of ubiquitinated proteins in marizomib cortical extracts (Fig. EV5A). To investigate the proteome-wide effect of proteasome inhibition, we performed a 10-plex TMT on cortical extracts from marizomib and vehicle cohorts (Fig. EV5B). With proteasome inhibition, there should be an accumulation proteins as they cannot be degraded. Quantification of protein abundance differences confirmed a significant accumulation with marizomib, as shown by the shift of the volcano plot and the number of proteins with significantly increased abundance (red)

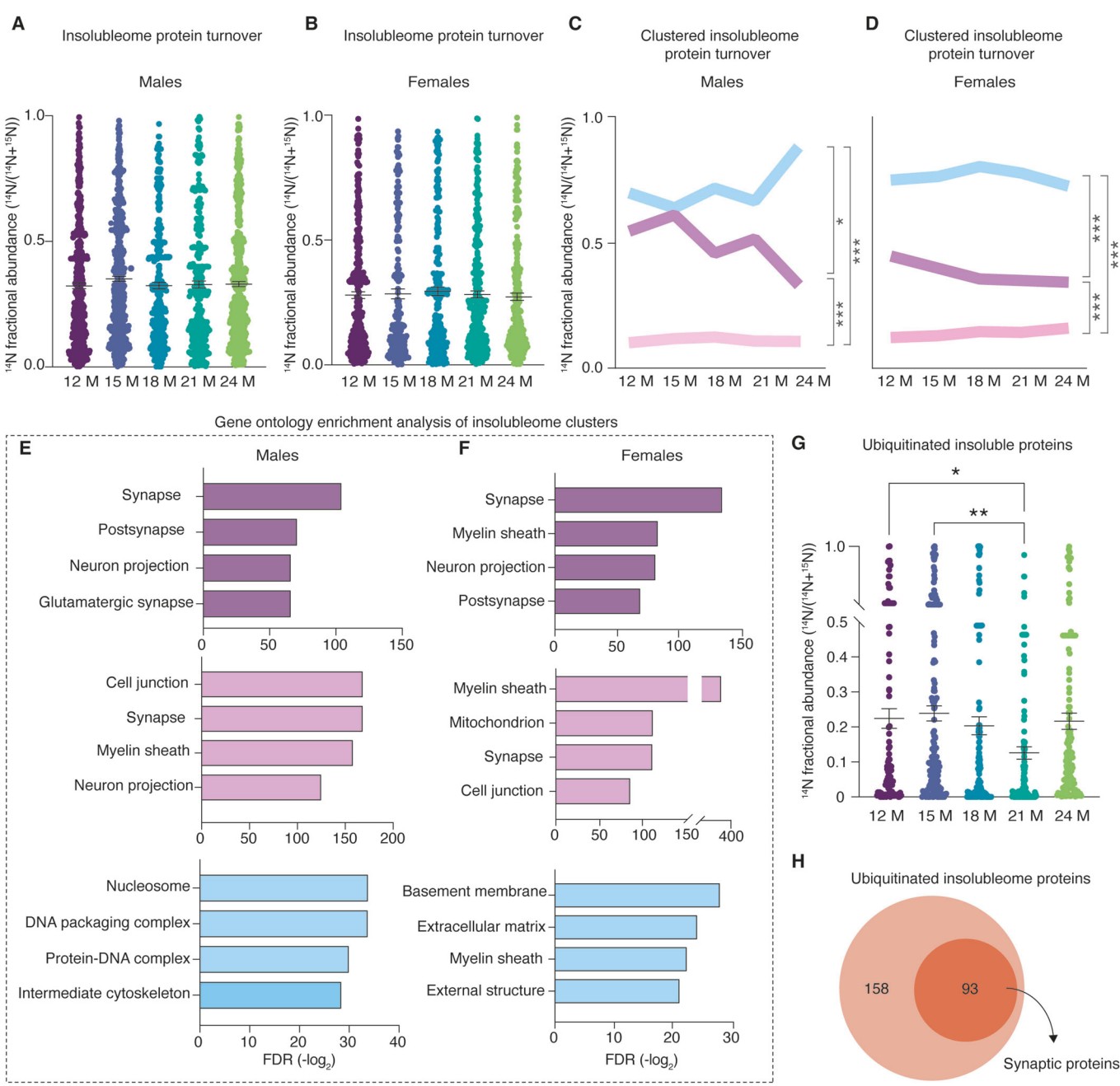

**Figure 3.   Protein misfolding contributes to differential protein turnover trends in the cortex.**

A, B Global $^{14}$N FA of insoluble proteome in male and female groups across aging, based on reconstructed MS1 chromatograms. Proteins identified as $^{14}$N or $^{15}$N peptides are included and quantified based on reconstructed MS1 chromatograms. C, D k-means clustering of $^{14}$N FA of male and female datasets revealed distinct protein turnover trends across aging. Blue, purple, and pink lines represent mean of each cluster. E, F Gene ontology enrichment analysis of respective trends revealed cell component terms significantly enriched. supramol = supramolecular. Purple, pink, and blue bar graphs represent the GO:CC enrichment analysis from proteins corresponding to respective clusters. The top four terms were chosen based on combined FDR and fold enrichment scores. G $^{14}$N FA for ubiquitinated insoluble proteins across aging based on reconstructed MS1 chromatograms. Modified proteins identified by either $^{14}$N or $^{15}$N peptides are included and quantified based on reconstructed MS1 chromatograms. H Venn diagram showing the fraction of the ubiquitinated insoluble protein pool associated with synaptic GO:CC terms. All data are mean ± SEM with $n = 4$ mice for both sexes except $n = 3$ for female 15 M and 24 M groups. G, H: $n = 4$ for all male cohorts. *$p$ value < 0.05; **$p$ value < 0.01; ***$p$ value < 0.001 by Kruskal–Wallis ANOVA with Tukey's multiple comparisons test or 2-way ANOVA. For k-means analysis, k groups were chosen based on optimal silhouette score.

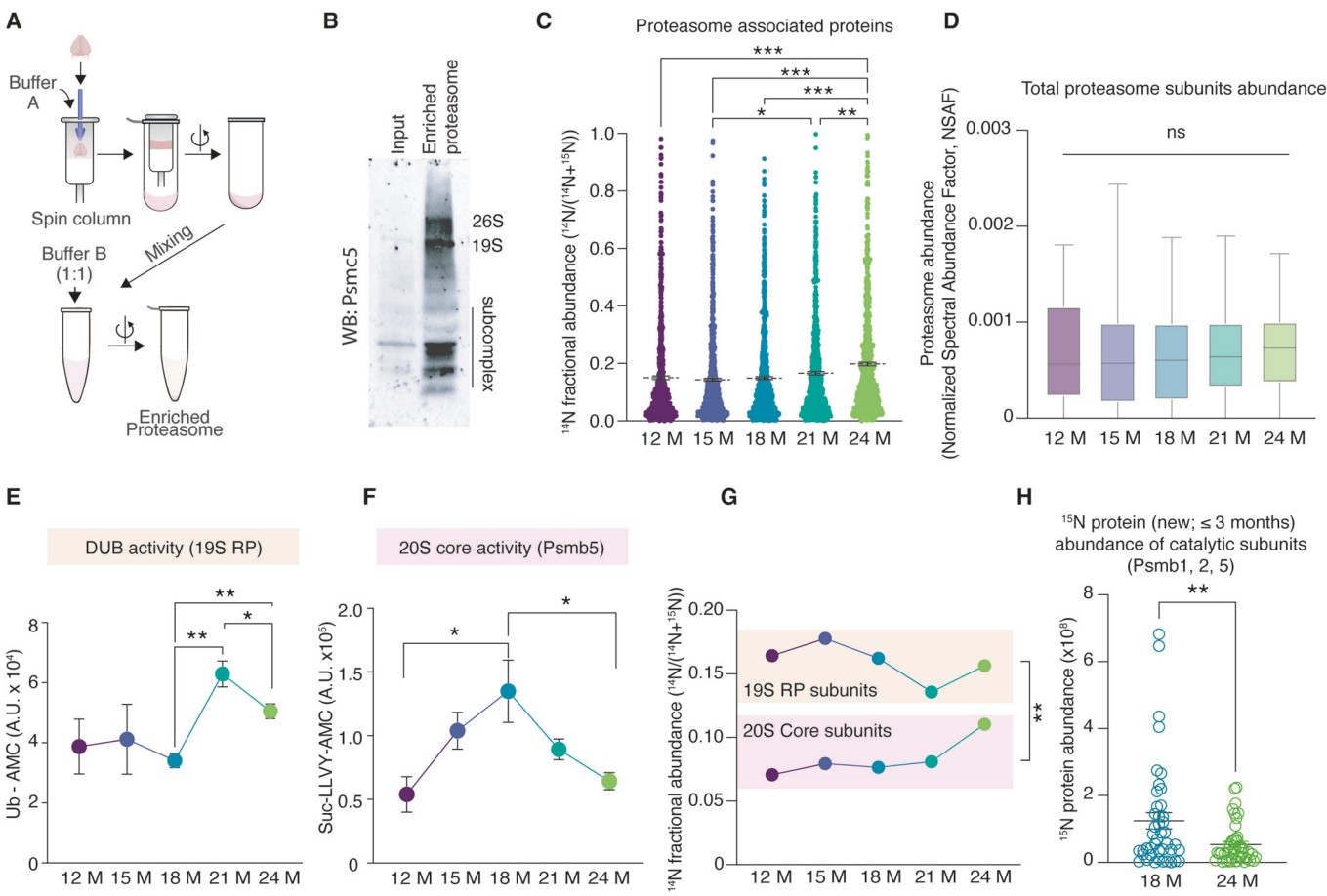

**Figure 4.  Age-associated fluctuations in proteasomal activity are tightly tied to catalytic subunit turnover.**

A Schematic depicting proteasome isolation from cortical extracts. **B** Western blot probing for Psmc5 showing enrichment of proteasome components. **C** Quantification of $^{14}$N FA of proteasome subunits and copurifying proteins across age groups reveals significant age-dependent linear decrease in proteasome efficiency. Proteins identified from either $^{14}$N or $^{15}$N peptides are included and quantified based on reconstructed MS1 chromatograms. **D** Protein abundance based on normalized spectral abundance factor (NSAF) values show no significant difference in overall changes in levels with aging. Box plot center is median with 25th to 75th percentile bounds. Whiskers extend from smallest to largest values. **E** Ub-AMC substrate activity assay with isolated proteasomes reveals deubiquitinase (DUB) activity of the 19 S regulatory particle (RP) is significantly different among age groups. **F** Suc-LLVY-AMC based Psmb5 20 S proteasome activity is significantly increased at 18 M compared to 12 M and 24 M. **G** 19 S RP has significantly higher $^{14}$N FA across aging compared to 20 S core proteasome subunits. **H** The catalytic proteasome core subunits (i.e., Psmb1, Psmb2, and Psmb5) have a significantly smaller pool of $^{15}$N (newly synthesized) protein at 24 M compared to 18 M. Each data point represents an individual quantified peptide. All data are mean ± SEM with $n = 4$ from the male cohorts. *$p$ value < 0.05; **$p$ value < 0.01; ***$p$ value < 0.001 by Kruskal–Wallis ANOVA with Tukey's multiple comparisons test, 2-way ANOVA, or Student's $t$ test. Source data are available online for this figure.

(Fig. 5B). GO enrichment confirmed that biological processes such as "protein folding" and "chaperon-mediated folding" were significantly overrepresented for this protein pool (Fig. EV5C). We next confirmed that pharmacological inhibition at 18 M modulates a common pool of proteins affected during normal aging. Over 100 proteins that significantly accumulated after marizomib were also proteasome-associated and showed slowed turnover during aging from 18 M to 21 M (Fisher's exact $p < 0.001$) (Fig. EV5D, Fig. 4D). This confirms that suppression of proteasome activity with marizomib affects many proteasome-associated proteins that experience age-associated decreases in turnover.

To assess the effects of marizomib on proteasome subunit turnover, we purified proteasomes from cortical extracts for activity

assays and MS analysis. There was no significant difference in the amount of isolated proteasomes (Fig. EV5E). Importantly, we confirmed that 20 S core activity is significantly decreased by marizomib, while DUB activity (19 S RP) remained unaffected (Fig. 5C,D). $^{12}$C6 FA - a metric of relative protein turnover - was then calculated from reconstructed MS1 chromatograms to quantify the amount of $^{12}$C6 proteins remaining relative to the total pool of proteasome-associated proteins ($^{12}$C6 and $^{13}$C6-lysine). The shift in the volcano plot showing increased $^{12}$C6 FA with marizomib indicates that proteasome inhibition results in slowed turnover of proteasome-associated proteins (Fig. EV5F). Finally, we found that marizomib-based suppression of proteasome activity results in significantly slowed turnover of proteasome subunits

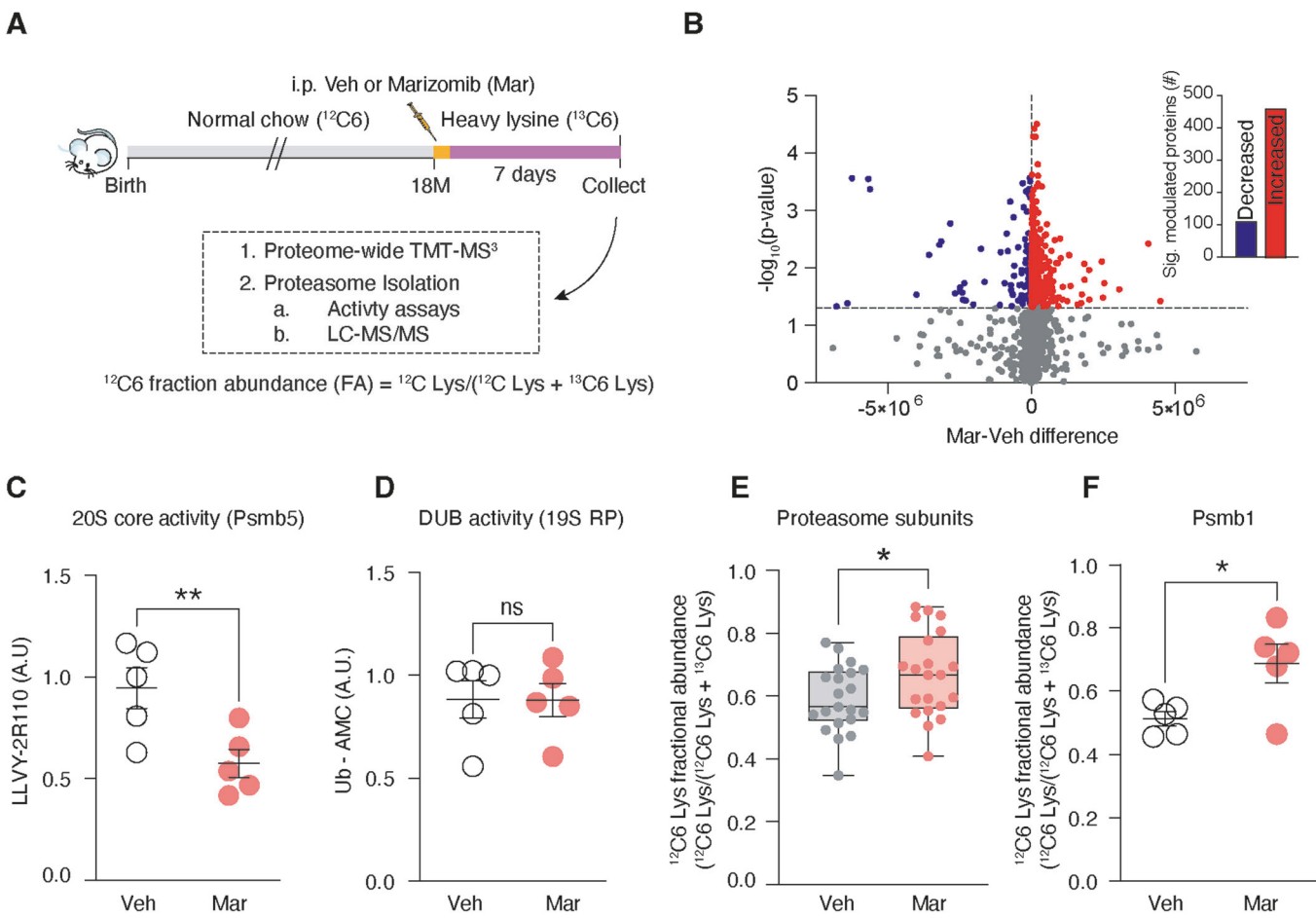

**Figure 5. Pharmacological suppression of 20 S activity at 18 M results in slowed turnover of proteasome subunits.**

A Experimental paradigm of pharmacological proteasome inhibition combined with in vivo stable isotope labeling. Marizomib (100 μg/kg) or vehicle (DMSO in saline) was administered to respective 18 M cohorts. 24 h after injection, both cohorts were provided exclusively $^{13}$C6-lysine chow. After 7 days, cohorts were collected for biochemical analyses. B Quantification of proteome-wide differences in abundance between marizomib and vehicle shows proteins significantly increased and decreased in red and blue, respectively. The number of modulated proteins (red = 462, blue = 112) is shown in inset. Total $n = 2433$ quantified proteins. C Suc-LLVY-2R110 based 20 S core (Psmb5) activity is significantly decreased with marizomib compared to vehicle. D Ub-AMC based DUB activity (19 S RP) shows no difference between marizomib and vehicle. E Proteasome subunits ($n = 21$ proteins) have significantly higher $^{12}$C6-lysine fractional abundance in marizomib compared to vehicle groups. Average $^{12}$C6-lysine fractional abundance calculated for each subunit (identified in at least 3 biological replicates) is represented by data points. Box plot center is median with 25th to 75th percentile bounds. Whiskers extend from smallest to largest values. F Psmb1, catalytically active 20 S core subunit, has significantly higher $^{12}$C6 fractional abundance with marizomib compared to vehicle. All data are mean ± SEM with $n = 5$ per group. *$p$ value < 0.05; **$p$ value < 0.01; ***$p$ value < 0.001 by Student's $t$ test. Source data are available online for this figure.

(Fig. 5E,F, Fig. EV5G). Taken together, pharmacological inhibition of proteasome activity results in slowed turnover of proteasome subunits and accumulation of proteins. This finding recapitulates our previously discovered relationship between proteasome subunit fidelity and activity during aging.

## Discussion

Despite the established link between proteostasis collapse and aging, our knowledge of changes in protein turnover dynamics during this process remains incomplete (Lopez-Otin et al, 2023). In this study, we combined continuous in vivo stable isotope metabolic labeling with quantitative proteomics to investigate

changes in protein turnover along the aging continuum with robust temporal resolution. Taken together, we found that brain protein turnover during aging is a highly dynamic process that varies between sexes and involves multiple cellular compartments. Misfolded and insoluble proteins, at least in part, underlie the fluctuating trends in protein turnover. Importantly, our results strongly suggest that age-related changes in proteasome activity are involved in the accumulation of ubiquitinated insoluble proteins and may result from altered turnover of the 20 S proteasomal catalytic subunits.

Our robust labeling approach captures changes that occur across multiple life stages, from early adulthood through adulthood and into old age in wild-type mice (Palliyaguru et al, 2021; Yanai and Endo 2021). Protein turnover rates vary by tissue type

(Bomba-Warczak et al, 2021; Fornasiero et al, 2018; Savas et al, 2012). Previous studies have used short-term labeling with amino acids highly enriched in heavy isotopes as a useful tool to determine protein lifetimes (Fornasiero et al, 2018; Kluever et al, 2022). The use of this technique is well suited for turnover changes that occur over short periods of time; however, studies of the turnover for long-lived protein that persist for months require longer labeling periods. Using this technique, we found that throughout the aging process, brain tissue contains the largest fraction of old proteins (≥3 months) (Fig. 1). Age-related fluctuations in protein turnover were unique to brain tissue, which was further confirmed by tracking the raw Mdh2 MS1 peptide spectra at each age. Long-lived protein pools, such as those in post-mitotic cells, may be particularly susceptible to accumulating damage over time. Therefore, the proteostasis systems in these cell types require tight regulation and highly efficient protein degradation machinery during aging. With the identification of this unique age-related phenomenon, our goal was to create a robust atlas of protein turnover dynamics in the brain over the course of aging.

Deep profiling of protein turnover dynamics during aging using TMT-MS analysis revealed pronounced sex-specific differences. In both sexes, we discovered a striking non-linear pattern of turnover, which is supported by previous observations that suggest aging occurs in waves (Piehl et al, 2022). Interestingly, the profile of global turnover was shifted later by nine months in females (Fig. 2). We speculated that this may indicate a slower aging process in female mice compared to males. Our results are consistent with previous reports that females have a longer lifespan and therefore would experience the same changes in turnover at a later timepoint (Palliyaguru et al, 2021; Yanai and Endo 2021).

In addition to sex-specific differences, we found that protein turnover trends during aging were associated with specific cellular compartments. Gene ontology enrichment analysis revealed that subcellular compartments, such as synapse and mitochondria, showed the earliest slowing of protein turnover in both males and females. This finding correlates with previous work showing that the same cellular compartments in the brain exhibit impaired protein turnover in early stages of AD (Hark et al, 2021). Remarkably, clusters between males and females showed striking similarities in GO:CC analyses despite temporally distinct profiles. For example, cluster 3 in males and cluster 1 in females showed shifted turnover trends, but the GO terms overrepresented in both, such as synapse, mitochondria, and myelin, remained the same. Previous work has shown age-dependent changes in myelin fragmentation and deposition intracellular into insoluble lysosomal inclusions (Safaiyan et al, 2016). In addition, many common proteins were found in the two different clusters, such as: Amph, Nefl, Nefh and Vat1. This indicated that despite sex-specific age differences in turnover, similar groups of proteins exhibited common behavior over time. This further suggested that there may be specific cellular regions of vulnerability that are differentially affected by aging.

This led us to explore if these distinct trends in protein turnover were driven by insoluble or misfolded proteins. This analysis identified distinct groups: (1) proteins with the highest $^{14}$N FA across age intervals (blue cluster), and (2) proteins with lower FA during aging (purple and pink clusters) (Fig. 3). Notably, the purple and pink clusters were strikingly overrepresented for common GO:CC terms related to synapse. The separation of clusters with a higher and lower $^{14}$N FA may represent proteins that misfold in an age-independent and age-dependent manner, respectively. Furthermore, we found that the different groups in both male and female insolubleomes possessed common affected cellular compartments and proteins, again suggesting that aging is temporally different between the sexes, but the proteomes behave similarly. Synaptic proteins such as Snap25, Syt1, Picalm, Stx1b, Syn1, and Sv2b were common to both male and female pink and purple clusters. In addition, we did not identify many mitochondrial terms in the insolubleome analysis, despite our previous finding that this subcellular compartment had a similar turnover trend as synaptic proteins. This suggests that synaptic proteins appear to be selectively prone to phase separation or misfolding, which may underlie protein turnover fluctuations during aging. However, this finding also showed that not all fluctuations in protein turnover could be explained by this phenomenon. Next, we discovered that synapse-associated proteins comprised a majority of the ubiquitinated insoluble protein pool. This striking observation suggested that these proteins may have been marked for degradation but were not degraded, resulting in insoluble accumulation. These results strongly suggest that these distinct insoluble proteins may have experienced UPS deficits leading to differential protein turnover during aging.

Isolation of intact proteasomes from cortical extracts allowed us to investigate whether UPS deficits lead to age-related fluctuations in turnover (Fig. 4). As several previous studies have shown that proteasomal activity decreases with age, we sought to investigate whether this could be a result of altered proteolytic activity, as opposed to abundance (Shen et al, 2021; Vecchi et al, 2020). Notably, this analysis confirmed a striking relationship between protein turnover and proteolytic activity. When 19 S RP and 20 S activity was at its highest, at 21 M and 18 M respectively, $^{14}$N FA for the corresponding proteasomal subunits was the lowest during aging. Conversely, when 19 S RP and 20 S core activity was low, the $^{14}$N FA was higher. This correlation in both directions strongly suggests that proteasome activity during aging is directly related to subunit renewal. Our pharmacological suppression of proteasome activity in combination with stable-isotope labeling allowed for direct assessment of the relationship between proteasome activity and turnover (Fig. 5). In this paradigm, $^{13}$C6-lysine was used to quantify turnover over the short 7-day period following proteasome inhibition. To mimic the decreased 20 S core activity observed during aging (from 18 M to 24 M), we administered marizomib at 18 M and examined whether proteasome subunit turnover was slowed as a result. Suppression of activity at 18 M resulted in slowed protein turnover of proteasome subunits, including the catalytically active subunit Psmb1—a recapitulation of the phenomenon observed during aging. We propose that the age-associated changes in proteasome subunit turnover in the brain lead to alterations in proteasome activity, resulting in downstream waves of protein misfolding and cellular compartment-specific turnover fluctuations.

In summary, our work reveals fluxes in brain protein turnover along the aging continuum that may underlie subcellular sites of vulnerability in age-related neurodegenerative diseases. We have also discovered sex-specific differences in protein turnover that are

unexpectedly linked to shared cellular compartments. Misfolding of distinct protein pools drives the turnover dynamics of selected cellular compartments, which may result from altered proteolytic activity. Finally, during aging, 19 S RP and 20 S core proteasome activity is strongly correlated with the turnover of their respective subunits, a phenomenon recapitulated with direct pharmacological modulation of proteasome activity. Collectively, our study highlights the impact of protein turnover dynamics during aging. Future studies focusing on improving proteasome fidelity by renewing catalytically active subunits may help to maintain a balanced protein turnover in the brain during aging.

# Methods

## Reagents and tools

See Table 1.

## Animals

All experiments performed were approved by the Institutional Animal Care and Use Committee of Northwestern University (Protocols IS0009900 and IS00010858 and IS00022178). C57BL/J6 mice were used for all experiments. These mice were originally obtained from the Jackson Laboratory. For euthanasia, mice were anesthetized with isoflurane, followed by cervical dislocation and acute decapitation. Male ($n = 4$–$5$) and female ($n = 3$ or $4$) mice were used for all experiments.

## Pulse-chase metabolic labeling in mice (SILAM)

The general method for labeling mice with $^{15}N$ isotope was described previously (Bomba-Warczak et al, 2021; Hark et al, 2021; Savas et al, 2012). In brief, wild type mice were fed Spirulina-based chow enriched in $^{15}N$ (Cambridge Isotopes Laboratories) for three months during the following time points: 9–12 M, 12–15 M,

**Table 1. Reagents and tools.**

| Reagent/Resource | Reference or Source | Identifier or Catalog Number |
|---|---|---|
| Experimental Models | | |
| C57BL/6 J (*M. musculus*) | Jackson Lab | Strain #000664 |
| Antibodies | | |
| Mouse anti-ubiquitin | Santa Cruz | Cat #sc-8017 |
| Mouse anti-actin | Santa Cruz | Cat #sc-8432 |
| Chemicals, Enzymes and other reagents | | |
| TMT11-131C Label Reagent | Thermo Fisher Scientific | Cat #A34807 |
| C18 Spin Columns | Thermo Fisher Scientific | Cat# 8970 |
| HyperSep™ C18 Cartridges | Thermo Fisher Scientific | Cat# 60108–302 |
| TEAB buffer | Thermo Fisher Scientific | Cat #90114 |
| LysC, MS Grade | Promega | Cat #VA1170 |
| Trypsin Gold, MS Grade | Promega | Cat #V5280 |
| ProteaseMAX | Promega | Cat #V2072 |
| Suc-LLVY-AMC | AAT Bioquest | Cat #13453 |
| Suc-LLVY-2R110 | AAT Bioquest | Cat #13451 |
| Ubiquitin-AMC | Life Sensors | Cat #SI220 |
| Marizomib | MedChem Express | Cat #HY-10985 |
| Software | | |
| Integrated Proteomics Pipeline-IP2 | http://www.integratedproteomics.com/ | |
| RawExtract 1.9.9 | http://fields.scripps.edu/downloads.php | |
| ShinyGO 0.76.3 | http://bioinformatics.sdstate.edu/go/ | |
| Graphpad Prism 9.4.1 | https://www.graphpad.com/ | |
| Orange Data Mining | https://orangedatamining.com/ | |
| Other | | |
| BCA assay kit | Thermo Fisher Scientific | Cat #23225 |
| Micro BCA assay kit | Thermo Fisher Scientific | Cat #23235 |
| TMT10plex™ Isobaric Label Reagents | Thermo Fisher Scientific | Cat #90111 |
| High pH Reversed-Phase Peptide Fractionation Kit | Thermo Fisher Scientific | Cat# 84868 |
| Minute™ Cytosolic Proteasome Enrichment Kit | Invent Biotechnologies Inc. | Cat #PT-040 |

15–18 M, 18–21 M, and 21–24 M. The $^{14}$N and $^{15}$N protein enrichment was calculated based on the shapes of the peptide isotope envelopes and reconstructed peak areas (MacCoss 2005).

## $^{13}$C6-Lysine labeling in mice with Marizomib

Marizomib (MedChemExpress) was dissolved 10% DMSO in saline (0.9% sodium chloride). In brief, equal numbers of 18 M male mice were randomly assigned to vehicle or marizomib groups. Experimenters were not blinded to groups. Cohorts were given single 100 μg/kg intraperitoneal injection of marizomib or vehicle (10% DMSO in Saline). 24 h after injection, mice were provided heavy $^{13}$C6-lysine chow for 7 days. The general method for labeling mice with $^{13}$C6-lysine isotope was described previously (Kluever et al, 2022). The $^{12}$C6 and $^{13}$C6 lysine protein enrichment was calculated based on the shapes of the peptide isotope envelopes and reconstructed peak areas (MacCoss 2005).

## LC-MS/MS sample preparation from tissues extracts

Samples from brain cortex, heart, and liver extracts were prepared for LC-MS/MS analysis as previously described (Hark et al, 2021; Rao and Savas 2021). In short, proteins were precipitated using methanol/chloroform precipitation, denatured with 8 M urea, and subsequently processed with ProteaseMAX, following the manufacturer's instructions (Promega, Cat# V2072). The samples were reduced with 5 mM Tris(2-carboxyethyl)phosphine (TCEP) at room temperature (RT), alkylated in the dark with 10 mM iodoacetamide (IAA), then diluted with 50 mM ammonium bicarbonate (ABC) and quenched with 25 mM TCEP. Then the samples were digested with sequencing grade modified trypsin (Promega, Cat# V5280) overnight at 37 °C. The reaction was subsequently stopped by acidification with 1% formic acid (FA); desalted and dried down with vacuum centrifugation for future resuspension and LC-MS/MS analysis.

## TMT- MS sample preparation from cortical extracts

TMT-MS sample preparation was performed as previously described (Rao and Savas 2021). In brief, homogenized cortical extracts were prepared and 200 μg of protein was precipitated using methanol-chloroform precipitation to separate proteins from lipids and impurities. Extracted protein was then resuspended in 6 M guanidine in (100 mM TEAB). Protein samples were further reduced of disulfide bonds with DTT, followed by alkylation of cysteine residues with IAA. Proteins were then digested for 3 h at RT with 1 μg LysC (Promega) and then digested overnight at 37 °C with 2 μg of Trypsin. The digest was then acidified with formic acid and desalted using C18 HyperSep columns (ThermoFisher Scientific). The eluted peptide solution was dried before resuspension in 100 mM HEPES. Micro BCA assay was subsequently performed to determine the concentration of peptides. 100 μg of peptide from each sample was then used for isobaric TMT labeling, following manufacturer's instructions (ThermoFisher Scientific). After incubating for 60 min., at room temperature; the reaction was quenched with 5% (v/v) hydroxylamine to 0.3%. Isobarically labeled samples were then combined 1:1:1:1:1:1:1:1:1:1:1 and quickly desalted with C18 HyperSep column. The combined TMT labeled peptide samples were fractionated into 8 fractions using high pH reversed-Phase columns (Pierce). Peptide solutions were dried, stored at −80 °C, and reconstituted in LC-MS Buffer A (5% acetonitrile, 0.125% formic acid) for LC-MS/MS analysis.

## Insolubleome sample preparation from cortical extracts

Homogenized cortical homogenates were sonicated and incubated with 1% SDS and followed by ultracentrifugation at $100,000 \times g$ at 4 °C for 60 min. The pellet was saved and supernatant was centrifuged again at $150,000 \times g$ at 4 °C for 60 min. The second pellet was dissolved in solubilization buffer (100 mM HEPES) and combined with the first pellet. All the SDS insoluble samples were precipitated, digested and prepared for MS analysis. A small fraction of each sample was saved for other analyses. Biophysical amyloid properties were calculated based on CamSol intrinsic (https://www-cohsoftware.ch.cam.ac.uk//index.php) and PASTA 2.0 (http://www.old.protein.bio.unipd.it/pasta2/) prediction tools.

## Proteasome isolation

The Minute$^{TM}$ Cytosolic Proteasome Enrichment Kit (Cat #PT-040) was utilized for proteasome enrichment following manufacturer's protocol. In short, 30 μg frozen tissue was ground using pestle and homogenized in Buffer A followed by filtration in the cartridge provided with the kit. Next, the collection tube was centrifuged at $16000 \times g$ for 30 min and supernatant was mixed 1:1 with Buffer B. After 10 min incubation on ice, and invert mix, tube was centrifuged at $10000 \times g$ for 10 min. The pellet was dissolved in cold PBS and saved for further analysis.

## Proteasome activity assays

To quantify 19 S RP proteasome activity, ubiquitin C-terminus derivative containing 7-amido-4-methylcoumarin (Ub-AMC), a fluorogenic substrate was utilized to monitor the ubiquitin hydrolase (deubiquitinase) activity. To quantify 20 S core proteasome activity, Suc-LLVY-AMC or Suc-LLVY-2R110, fluorogenic substrates were utilized to monitor the chymotrypsin-like activity. Enriched proteasome samples were quantified using μBCA colorimetric assay and 0.25-1 mg protein samples were incubated for 30 min with 20 μM of respective substrates. The endpoint fluorescence intensity was measured at 460 nm (excitation at 360 nm). Appropriate experimental controls were included for background intensity reduction. Each experiment included 4-5 biological replicates with two technical replicates.

## LC-MS/MS quantification of protein turnover

Dried samples were resuspended in 20 μl Buffer A (94.875% H2O with 5% ACN and 0.125% FA) and three micrograms, as determined by microBCA assay (Thermo Scientific, Cat# 23235) of each fraction or sample were loaded via auto-sampler with a Thermo EASY nLC 100 UPLC pump onto a vented Pepmap 100, 75 μm × 2 cm, nanoViper trap column coupled to a nanoViper analytical column (Thermo Scientific) with stainless steel emitter tip assembled on the Nanospray Flex Ion Source with a spray voltage of 2000 V. A coupled Orbitrap Fusion was used to generate MS data. Buffer A contained 94.785% H2O with 5% ACN and 0.125% FA, and buffer B contained 99.875 ACN with 0.125% FA. The chromatographic run was 4.5 h in total with the following

profile of Buffer B: 2% for 7 min, 2–7% for 1 min, 7–10% for 5 min, 10–25% for 160 min, 25–33% for 40 min, 33–50% for 7 min, 50–95% for 5 min, 95% for 15 min, then back to 2% for the remaining 30 min. For quantification of diGly (K-GG) ubiquitin remnant from insolubleome LC-MS/MS experiments, a search parameter using a differential modification of 114.042927 CKST was used to restrict analysis to DiGly modified peptides. Peptides from each biological replicated for respective groups was pooled for analysis.

## TMT-MS data collection

TMT-MS[3] analysis was performed as previously described (Rao and Savas [2021]). In short, samples were resuspended in 20 µl Buffer A (5% acetonitrile, 0.125% formic acid) and micro-BCA was performed. 3 µg of each fraction was loaded for LC-MS analysis via an auto-sampler with a Thermo EASY nLC 100 UPLC pump onto a vented Pepmap100, 75 µm × 2 cm, nanoViper trap column coupled to a nanoViper analytical column (Thermo Scientific) with stainless steel emitter tip assembled on the Nanospray Flex Ion Source with a spray voltage of 2000V. Orbitrap Fusion was used to generate MS data. The chromatographic run was performed with a 4 h gradient beginning with 100% Buffer A and 0% B and increased to 7% B over 5 min, then to 25% B over 160 min, 36% B over 40 min, 45% B over 10 min, 95% B over 10 min, and held at 95% B for 15 min before terminating the scan. Buffer A contained 5% ACN and 0.125% formic acid in H20, and Buffer B contained 99.875 ACN with 0.125% formic acid. Multinotch MS3 method was programmed as the following parameter: Ion transfer tube temp = 300 °C, Easy-IC internal mass calibration, default charge state = 2 and cycle time = 3 s. MS1 detector set to orbitrap with 60 K resolution, wide quad isolation, mass range = normal, scan range = 300–1800 m/z, max injection time = 50 ms, AGC target = $2 \times 105$, microscans = 1, RF lens = 60%, without source fragmentation, and datatype = positive and centroid. Monoisotopic precursor selection was set to included charge states 2–7 and reject unassigned. Dynamic exclusion was allowed n = 1 exclusion for 60 s with 10ppm tolerance for high and low. An intensity threshold was set to $5 \times 103$. Precursor selection decision = most intense, top speed, 3 s. MS2 settings include isolation window = 0.7, scan range = auto normal, collision energy = 35% CID, scan rate = turbo, max injection time = 50 ms, AGC target = $6 * 10^5$, Q = 0.25. In MS3, the top ten precursor peptides were selected for analysis (i.e., SPS-MS3) were then fragmented using 65% HCD before orbitrap detection. A precursor selection range of 400–1200 m/z was chosen with mass range tolerance. An exclusion mass width was set to 18 ppm on the low and 5 ppm on the high. Isobaric tag loss exclusion was set to TMT reagent. Additional MS3 settings include an isolation window = 2, orbitrap resolution = 60 K, scan range = 120–500 m/z, AGC target = $6*10^5$, max injection time = 120 ms, microscans = 1, and datatype = profile.

## TMT-MS data analysis and quantification

TMT-MS[3] data analysis was performed as previously described with quantitative changes to calculate protein turnover with TMT-MS (Rao and Savas [2021]). In short, protein identification, TMT quantification, and analysis were performed with The Integrated Proteomics Pipeline-IP2 (Integrated Proteomics Applications, Inc.,

http://www.integratedproteomics.com/). Proteomic results were analyzed with ProLuCID, DTASelect2, Census, and QuantCompare. MS1, MS2, and MS3 spectrum raw files were extracted using RawExtract 1.9.9 software (http://fields.scripps.edu/downloads.php). Pooled spectral files from all eight fractions for each sample were then searched against the Uniprot mouse protein database and matched to sequences using the ProLuCID/SEQUEST algorithm (ProLuCID ver. 3.1) with 50 ppm peptide mass tolerance for precursor ions and 600 ppm for fragment ions. Fully and half-tryptic peptide candidates were included in search space, all that fell within the mass tolerance window with no miscleavage constraint, assembled and filtered with DTASelect2 (ver. 2.1.3) through the Integrated Proteomics Pipeline (IP2 v.5.0.1, Integrated Proteomics Applications, Inc., CA, USA). Static modifications at 57.02146 C and 229.1629 K at N-term were included. The target-decoy strategy was used to verify peptide probabilities and false discovery ratios (McAlister et al, [2014]). Minimum peptide length of five was set for the process of each protein identification and each dataset included a 1% FDR rate at the protein level based on the target-decoy strategy. Isobaric labeling analysis was established with Census 2 as previously described TMT channels were normalized by dividing it over the sum of all channels and with a bridge channel to compare between TMT-MS experiments (McAlister et al, [2014]; Rao and Savas [2021]). No intensity threshold was applied. DTA selection of the pools of [14]N and [15]N proteins was separated before performing TMT quantification. TMT intensities for each pool were then used to represent [14]N and [15]N abundances and for calculation of protein turnover.

## Gene ontology enrichment analysis

For Gene ontology analysis, we used protein analysis through evolutionary relationships (ShinyGo) for cellular component categories (http://bioinformatics.sdstate.edu/go/). The statistical overrepresentation test was calculated by using the significant proteins identified from respective clusters as the query and the aggregated total proteins identified in all groups as the reference. Ontologies with Fisher statistical tests with false discovery rate (FRD) correction less than .05 were considered significant. Protein characterizing the intrinsic biophysical properties of proteins, we calculated individual scores for overall solubility and best energy for folding using CamSol Intrinsic (https://www-cohsoftware.ch.cam.ac.uk//index.php) and PASTA 2.0 (http://old.protein.bio.unipd.it/pasta2/) online servers. We obtained FASTA sequences for each protein identified in our clustering analysis and used default parameters in both servers to plot calculated solubility and energy scores.

## Western blotting

Protein samples were subjected to BCA assay for quantification and normalization. 20 µg of each sample was then prepared for western blots by adding 6X SDS sample buffer. The mixtures were sonicated and boiled at 95 °C for 5 min each and then loaded in a 16% Tris-glycine gel. Gels were run at 80 V for 4 h, then were transferred (BioRad semidry transfer) to a 0.45 µm nitrocellulose membrane. Membranes were then blocked with blocking Buffer (5% nonfat milk) in TBST for 1 h then incubated overnight with primary antibodies. Next day, membranes were washed four times with

TBST and incubated in secondary HRP-conjugated antibodies for 1 h at RT. Following four washes, the blots were developed BioRad Chemidoc system using SuperSignal West Pico PLUS substrate solution.

## Bioinformatic and statistical analysis

Statistical analyses were performed using GraphPad Prism or Orange Data Mining platforms. All values in figures with error bars are presented as mean ± SEM. Comparisons across all five groups were compared by one-way ANOVA and post-hoc Tukey's test. Two-sided Student's t-test was used for appropriate direct comparisons. $p$ values < 0.05 were considered statistically significant. Hierarchical clusters was performed in Orange to identify clustering. The number of clusters (k) was selected based on optimal silhouette score and minimum 10 protein group size. PCA plots were performed on the FA of proteins identified in each biological replicate. Heat maps are scaled by row (z-score). All data are mean ± SEM. *$p$ value < 0.05; **$p$ value < 0.01; ***$p$ value < 0.001.

# Data availability

The mass spectrometry proteomics data have been deposited to the ProteomeXchange Consortium with the following identifier: PXD043023 and on the MassIVE repository with the identifier: MSV000092174.

# Peer review information

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

## Acknowledgements

This work was supported by NIH grants R01 AG078796, R21 AG080248, and R21 AG080705 to J.N.S. and F31 AG079653 and T32 AG20506 to N.R.R. We also sincerely thank Eugenio Fornasiero and members of the Savas lab for their constructive feedback on this manuscript.

## Author contributions

**Nalini R Rao**: Conceptualization; Data curation; Validation; Investigation; Methodology; Writing—original draft; Writing—review and editing. **Arun Upadhyay**: Conceptualization; Data curation; Validation; Methodology; Writing—original draft; Writing—review and editing. **Jeffrey N Savas**: Conceptualization; Resources; Data curation; Supervision; Investigation; Methodology; Writing—original draft; Project administration; Writing—review and editing.

## Disclosure and competing interests statement

The authors declares no competing interests.

# Expanded View Figures

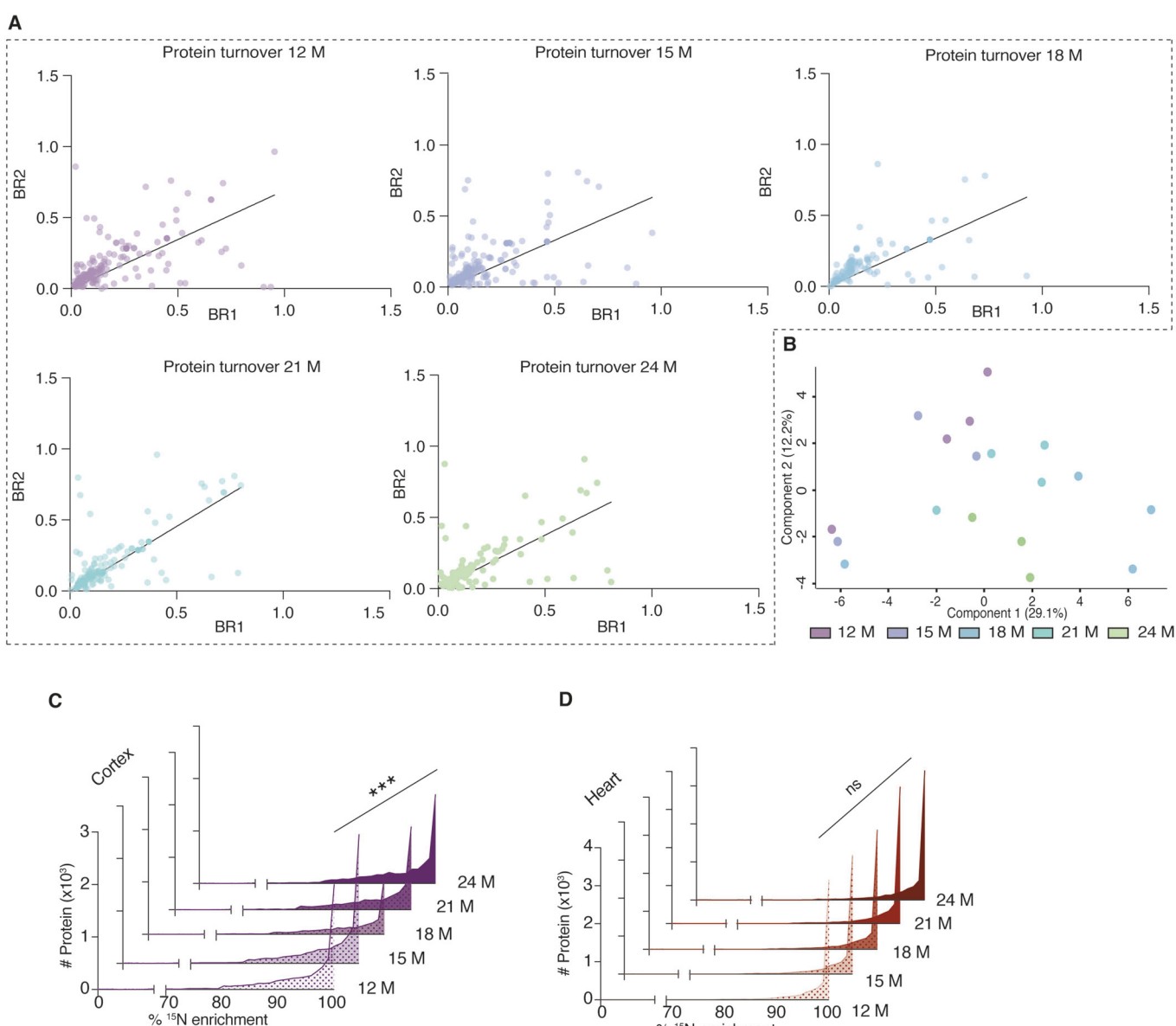

**Figure EV1.  ¹⁵N labeling efficiency and proteome-wide measures of ¹⁵N incorporation in cortex and heart.**

A Biological replicates from each age group show reproducible $^{14}N$ fractional abundance ($^{14}N/(^{14}N + ^{15}N)$) in cortex. B PCA analysis of each biological replicate from the 5 age groups. C, D $^{15}N$ incorporation plots for each age group in the cortex and heart showing the proportion of the proteome that was labeled with $^{15}N$. All data are mean ± SEM with $n = 3$–4 female mice. *$p$ value < 0.05; **$p$ value < 0.01; ***$p$ value < 0.001 by Kruskal–Wallis ANOVA with Tukey's multiple comparisons test.

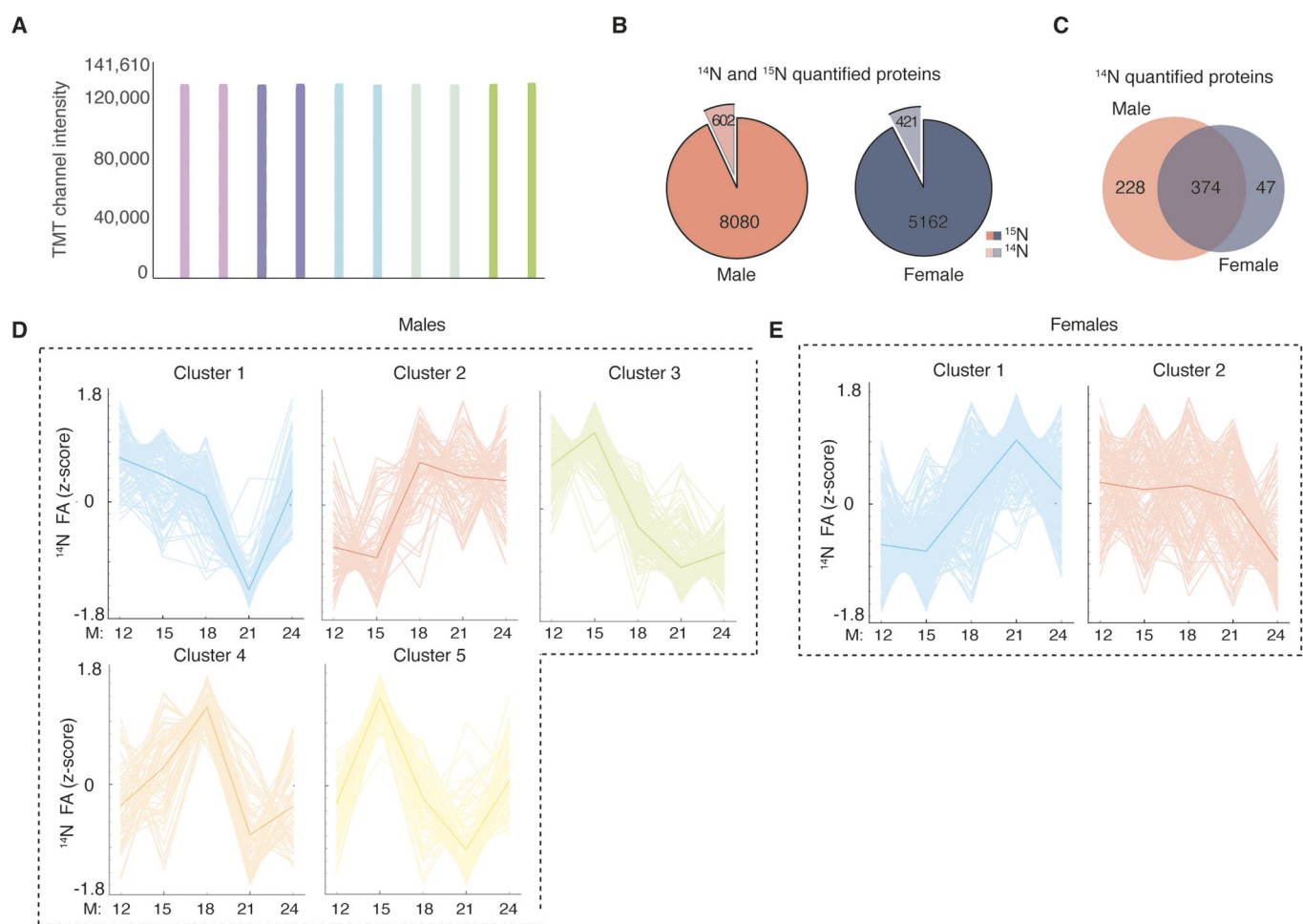

**Figure EV2. Quantification of proteins identified from TMT-MS experiments using novel analytical workflow.**

A Representative global TMT channel peak intensities for each reporter ion from 10-plex TMT-MS³ experiment demonstrating equal labeling across all channels. B Pie charts depicting quantified ¹⁴N and ¹⁵N proteins from TMT-MS³ analysis of male and female datasets. C Venn diagram showing the overlap between male and female quantified ¹⁴N proteins. D, E Line plots for each cluster from heatmap showing trends of protein turnover for males and females. All data are mean ± SEM with $n = 4$ mice for both sexes except $n = 3$ for female 15 M and 24 M groups.

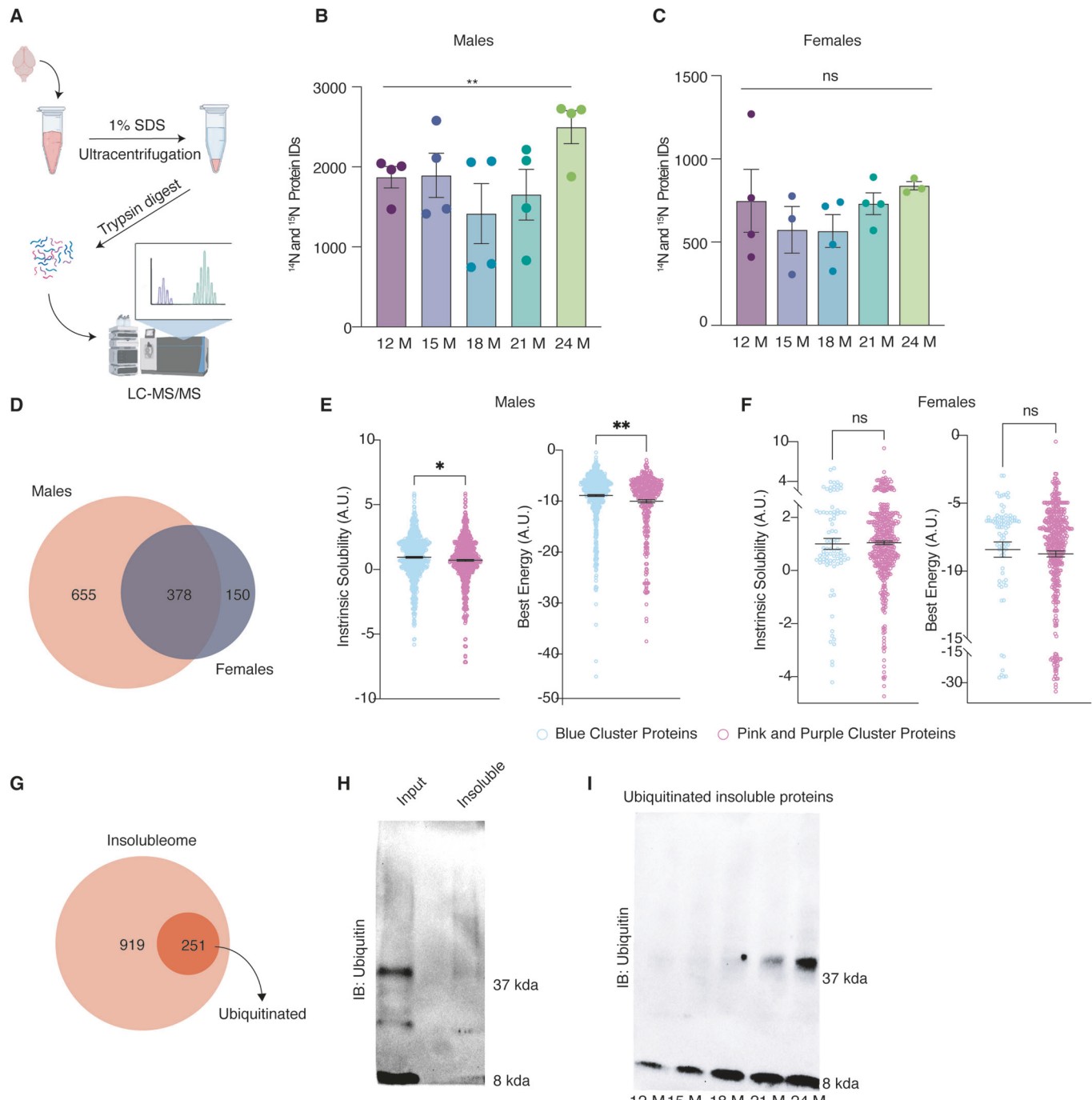

**Figure EV3. Confirmation of biochemical isolation and MS-based analysis of insoluble cortical proteome.**

A Schematic depicting the biochemical isolation and analytical workflow of the insoluble proteome from cortical extracts. B, C Number of protein identifications for each age group in male and female datasets. D Venn diagram showing overlap of identified insoluble proteins from male and female datasets. E, F Intrinsic solubility and best energy of purple and pink versus blue clusters from male and female datasets. G Venn diagram showing number of identified ubiquitinated proteins compared to insolubleome. H Western blot showing presence of ubiquitinated proteins in insoluble protein pools. I Western blot showing age-dependent increase in ubiquitinated proteins present in the insoluble pool in the indicated age groups in males. All data are mean ± SEM with $n = 4$ mice for both sexes except $n = 3$ for female 15 M and 24 M groups. [14]N FA is standardized to median of 12 M group. *$p$ value < 0.05; **$p$ value < 0.01; ***$p$ value < 0.001 by Kruskal–Wallis ANOVA with Tukey's multiple comparisons test or Student's $t$ test.

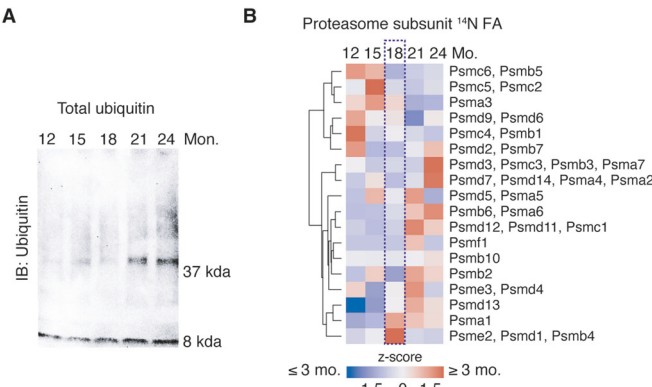

**A** Total ubiquitin

**B** Proteasome subsunit ¹⁴N FA

**Figure EV4.   Ubiquitination and proteasome core turnover profiles.**

A Western blot for ubiquitinated protein levels from isolated proteasomes shows increased abundance at 21 M and 24 M. B Heatmap representation of quantified proteasome subunit ¹⁴N fractional abundance across aging cohorts. All data are mean ± SEM with $n = 4$.

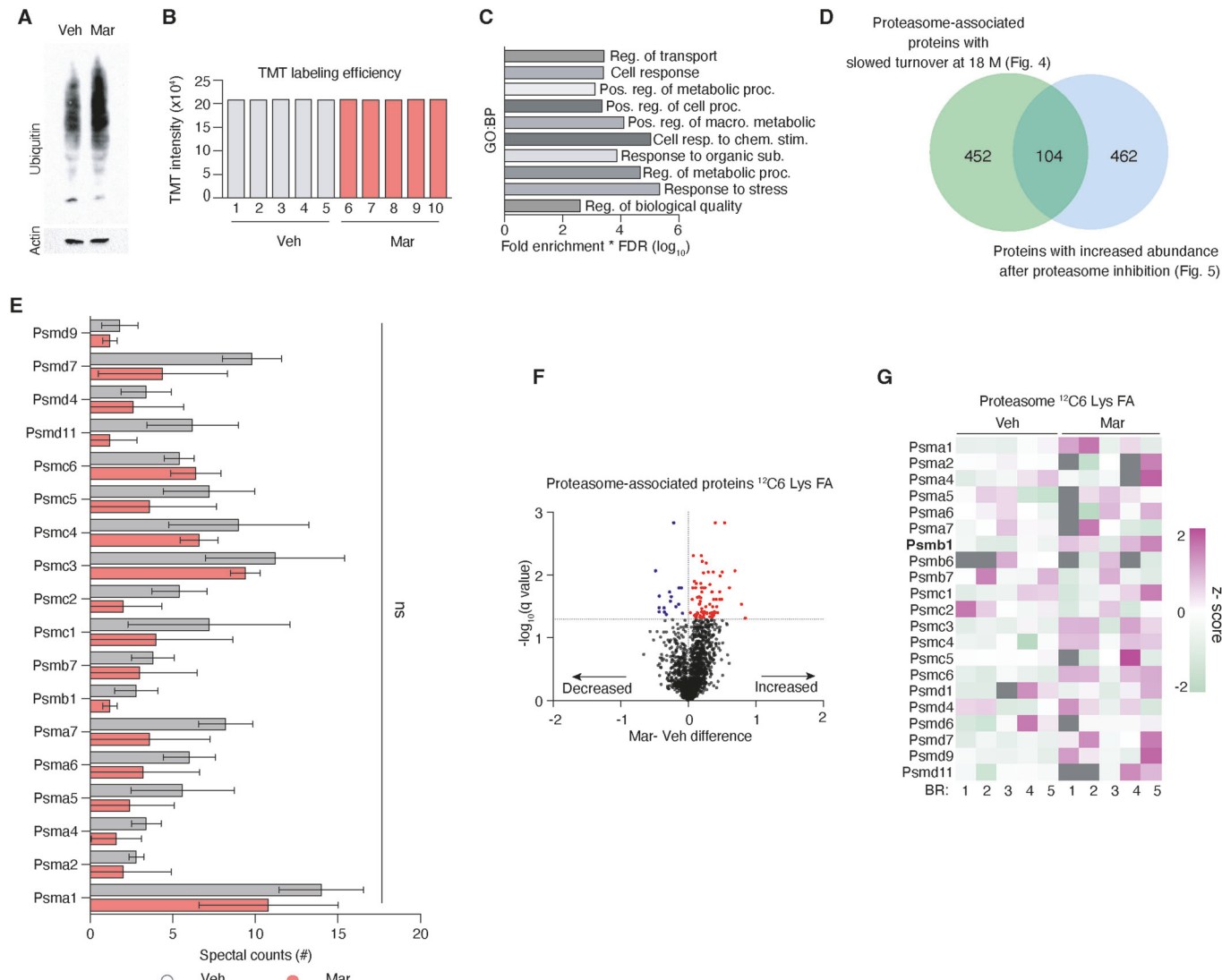

**Figure EV5. Detailed effects of partially suppressed proteasome activity with marizomib.**

A Western blot confirming accumulation of ubiquitinated proteins after marizomib treatment. B Representative global TMT channel peak intensities for each reporter ion from 10-plex TMT-MS³ experiment demonstrating equal labeling across all channels. C Gene ontology enrichment analysis showing the biological processes that are overrepresented in the group of proteins with significantly increased abundance after marizomib. D Confirmation that about one-fourth of proteasome subunits and copurifying proteins previously found modulated during aging are similarly modulated by marizomib. $p < 0.001$ by Fisher's exact test. E No differences in quantity proteasome subunits proteins identified between vehicle and marizomib cohorts after biochemical proteasome isolation. F Significant differences between marizomib and vehicle cohorts is show with red and blue representing increased and decreased ¹²C6-lysine fractional abundance, respectively. Total $n = 1426$ quantified proteins. G Heatmap representation of ¹²C6-lysine fractional abundance of quantified proteasome subunits show slowed turnover with marizomib. All data are mean ± SEM with $n = 5$ males. *$p$ value < 0.05; **$p$ value < 0.01; ***$p$ value < 0.001.

