## [Peer Review File · Molecular Systems Biology]

Derailed protein turnover in the aging mammalian brain.

Jeffrey Savas, Nalini Rao, and Arun Upadhyay

DOI: 10.15252/msb.202311808

Corresponding author(s): Jeffrey Savas (jeffrey.savas@northwestern.edu)

Review Timeline:

Submission Date:	5th Jun 23
Editorial Decision:	28th Jul 23
Revision Received:	2nd Nov 23
Editorial Decision:	8th Dec 23
Revision Received:	11th Dec 23
Accepted:	13th Dec 23

Editor: Maria Polychronidou

Transaction Report:

28th Jul 2023

Manuscript Number: MSB-2023-11808

Title: Derailed protein turnover in the aging mammalian brain.

Dear Dr Savas,

Thank you again for submitting your work to Molecular Systems Biology. We have now heard back from two of the three reviewers who agreed to evaluate your study. Unfortunately, even though we have sent them several reminders, we have not received a report from reviewer #2. In the interest of time, we have decided to proceed with making a decision based on the two available reports. As you will see below the two reviewers acknowledge that the presented data and findings seem interesting. They do however raise a series of concerns which we would ask you to address the issues raised in a major revision.

As you will see below most of the reviewers' comments refer to the need to provide further clarifications, include additional references and discuss some of the findings in further detail. A more fundamental issue is raised by reviewer #3 who recommends testing the emerging hypothesis that "reduced UPS activity after 18 months of age is responsible for the changes in the turnover of the catalytic core subunits of the proteasome observed at later ages" by treating 18-month old mice with proteasome inhibitors (e.g. bortezomib) and assessing the effect of the proteasome inhibition.

All issues raised by the reviewers need to be satisfactorily addressed. As you may already know, our editorial policy allows in principle a single round of major revision, so it is essential to provide responses to the reviewers' comments that are as complete as possible. Please feel free to contact me in case you would like to discuss in further detail any of the issues raised or if you would like to share your revision plan with me. I would be happy to schedule a call.

On a more editorial level, we would ask you to address the following points:

- Please provide a .doc version of the manuscript text (including legends for the main figures and EV Figures) and individual production quality figure files for the main Figures and EV Figures (one file per figure).
- We have replaced Supplementary Information by the Expanded View (EV format). In this case, all additional figures* can be provided as EV Figures. Please provide one file per EV Figure. Their legends should be included in the manuscript text. *If after the performed revisions the number of additional Figures is > 6, then they can all be included in a PDF called Appendix. Appendix figures should be labeled and called out as: "Appendix Figure S1, Appendix Figure S2... Appendix Table S1..." etc. Each legend should be below the corresponding Figure/Table in the Appendix. Please include a Table of Contents in the beginning of the Appendix. For detailed instructions regarding expanded view please refer to our Author Guidelines: .
- Supplementary Tables S1-S4 should be provided as Datasets EV1-EV4. Please provide one file per EV dataset. The description of each dataset should be included in the excel file, as a separate sheet.
- Please provide a "standfirst text" summarizing the study in one or two sentences (approximately 250 characters), three to four "bullet points" highlighting the main findings and a "synopsis image" (550px width and max 400px height, jpeg format) to highlight the paper on our homepage.
- All Materials and Methods need to be described in the main text. We would encourage you to use 'Structured Methods', our new Materials and Methods format. According to this format, the Materials and Methods section should include a Reagents and Tools Table (listing key reagents, experimental models, software and relevant equipment and including their sources and relevant identifiers) followed by a Methods and Protocols section in which we encourage the authors to describe their methods using a step-by-step protocol format with bullet points, to facilitate the adoption of the methodologies across labs. More information on how to adhere to this format as well as downloadable templates (.doc or .xls) for the Reagents and Tools Table can be found in our author guidelines: . An example of a Method paper with Structured Methods can be found here:
- Please include a "Disclosure and Competing Interests Statement" in the main text.
- Please include a Data availability section describing how the data, code etc. have been made available. This section needs to be formatted according to the example below:
The datasets and computer code produced in this study are available in the following databases:
 - Chip-Seq data: Gene Expression Omnibus GSE46748 (<https://www.ncbi.nlm.nih.gov/geo/query/acc.cgi?acc=GSE46748>)
 - Modeling computer scripts: GitHub (<https://github.com/SysBioChalmers/GECKO/releases/tag/v1.0>)
 - [data type]: [full name of the resource] [accession number/identifier] ([doi or URL or identifiers.org/DATABASE:ACCESSION])
- For data quantification: please specify the name of the statistical test used to generate error bars and P values, the number (n)

of independent experiments (specify technical or biological replicates) underlying each data point and the test used to calculate p-values in each figure legend. The figure legends should contain a basic description of n, P and the test applied. Graphs must include a description of the bars and the error bars (s.d., s.e.m.).

- The References should be formatted according to the Molecular Systems Biology reference style.

- When you resubmit your manuscript, please download our CHECKLIST (<https://bit.ly/EMBOPressAuthorChecklist>) and include the completed form in your submission.

Please note that the Author Checklist will be published alongside the paper as part of the transparent process (<https://www.embopress.org/page/journal/17444292/authorguide#transparentprocess>).

If you feel you can satisfactorily deal with these points and those listed by the referees, you may wish to submit a revised version of your manuscript. Please attach a covering letter giving details of the way in which you have handled each of the points raised by the referees. A revised manuscript will be once again subject to review and you probably understand that we can give you no guarantee at this stage that the eventual outcome will be favorable.

Kind regards,

Maria

Maria Polychronidou, PhD
Senior Editor
Molecular Systems Biology

We realize that it is difficult to revise to a specific deadline. In the interest of protecting the conceptual advance provided by the work, we recommend a revision within 3 months (26th Oct 2023). Please discuss the revision progress ahead of this time with the editor if you require more time to complete the revisions. Use the link below to submit your revision:

IMPORTANT: When you send your revision, we will require the following items:

1. the manuscript text in LaTeX, RTF or MS Word format
2. a letter with a detailed description of the changes made in response to the referees. Please specify clearly the exact places in the text (pages and paragraphs) where each change has been made in response to each specific comment given
3. three to four 'bullet points' highlighting the main findings of your study
4. a short 'blurb' text summarizing in two sentences the study (max. 250 characters)
5. a 'thumbnail image' (550px width and max 400px height, Illustrator, PowerPoint or jpeg format), which can be used as 'visual title' for the synopsis section of your paper.
6. Please include an author contributions statement after the Acknowledgements section (see <https://www.embopress.org/page/journal/17444292/authorguide>)
7. Please complete the CHECKLIST available at (<https://bit.ly/EMBOPressAuthorChecklist>). Please note that the Author Checklist will be published alongside the paper as part of the transparent process (<https://www.embopress.org/page/journal/17444292/authorguide#transparentprocess>).
8. When assembling figures, please refer to our figure preparation guideline in order to ensure proper formatting and readability in print as well as on screen: <https://bit.ly/EMBOPressFigurePreparationGuideline>
See also figure legend guidelines: <https://www.embopress.org/page/journal/17444292/authorguide#figureformat>
9. Please note that corresponding authors are required to supply an ORCID ID for their name upon submission of a revised manuscript (EMBO Press signed a joint statement to encourage ORCID adoption). (<https://www.embopress.org/page/journal/17444292/authorguide#editorialprocess>)
Currently, our records indicate that there is no ORCID associated with your account.

Please click the link below to provide an ORCID:

Link Not Available

The system will prompt you to fill in your funding and payment information. This will allow Wiley to send you a quote for the article processing charge (APC) in case of acceptance. This quote takes into account any reduction or fee waivers that you may be eligible for. Authors do not need to pay any fees before their manuscript is accepted and transferred to the publisher.

EMBO Press participates in many Publish and Read agreements that allow authors to publish Open Access with reduced/no publication charges. Check your eligibility: <https://authorservices.wiley.com/author-resources/Journal-Authors/open-access/affiliation-policies-payments/index.html>

*** PLEASE NOTE *** As part of the EMBO Press transparent editorial process initiative (see our Editorial at <https://dx.doi.org/10.1038/msb.2010.72>), Molecular Systems Biology publishes online a Review Process File with each accepted manuscripts. This file will be published in conjunction with your paper and will include the anonymous referee reports, your point-by-point response and all pertinent correspondence relating to the manuscript. If you do NOT want this File to be published, please inform the editorial office at msb@embo.org within 14 days upon receipt of the present letter.

Reviewer #1:

Rao et al. conducted continuous in vivo metabolic labeling to investigate age-related changes in protein turnover in mice. They compared three tissues - two postmitotic and one mitotic. As expected, the mitotic tissue exhibited greater label incorporation than the postmitotic ones. However, only the cortex displayed age-dependent differences. Intrigued by the findings in the cortex, the researchers delved deeper and performed additional measurements, this time comparing age-related protein turnover changes in males and females. Surprisingly, while proteins from the same cellular compartments exhibited shifts in turnover due to aging, the shifts in female mice occurred at later time points compared to male mice. Part of the age-related shift in protein dynamics could be attributed to the formation of protein aggregates that evade degradation. Interestingly, not all compartments that displayed age-related turnover changes were represented in these insolublesome. Furthermore, the proteasome, a critical player in protein degradation, exhibited differential activity throughout aging. The regulatory proteasome subunits (19S) demonstrated a distinct activity pattern from the core proteasome (20S).

Overall, while this study is descriptive in nature, it presents a fascinating dataset that will undoubtedly interest various scientific communities, including those focused on aging, neurodegeneration, and protein turnover. The data quality appears to be high, and although additional experiments may not be necessary in principle, certain terms and statements in the writing require further clarification. Addressing these points would be beneficial before acceptance.

1. Figure 1B and C and also the corresponding text in the results section (2nd page of results, 2nd paragraph): The authors use the term "protein abundance" here. This is a bit misleading as they mean number of identified proteins, while nowadays protein abundance means more protein amount of the proteins. The wording should be adjusted accordingly. This is also important in light of Figure 1D where not the number of IDs are used to calculate FA, but the integrated MS signals, which is more in line with an abundance definition in proteomics.
2. Figure 1B: Is the number of ID'd proteins in heart not significantly lower than in cortex or liver? Why was not the same cross-sample comparison performed as in 1C?
3. In Figure 1 was the data generated from female and male animals or only from one sex? Please, add the information to the legend. Same for Suppl. Figure 1.
4. Figure 2B and C: Relative 15N signal to WHAT? Is it to the median of 12 months as indicated at the bottom of the figure for 14N FA? If to the median of 12 months for each sex then why is the 12 months median for example at 2C not at a log₂ value of 0? This is throughout all the figures (see more below). Please define relative to WHAT as it is ambiguous sometimes. Moreover, in several figures it does not seem that it is true that it was standardized to the median of 12 months as then the median of 12months (assuming the line in the boxplot is the median) should be at 1 in linear space or 0 in log₂ space. I might be mistaken what you mean by "standardized to the median of 12 months", but in any case, a clarification would help.
5. Figure 2D, E: It says that "relative 14N protein abundance from both male and female cohorts NO significant differences..." I think the figure shows the opposite here - that there is a significant difference, doesn't it? Please correct.
6. Figure 2G: K-means clustering. Honestly, I see here more than 2 clusters for the female data. Please explain the rationale, why only 2 clusters were chosen as parameter for the female mice?
7. Figures 3 A, B and C, D - based on the y-axis label the plots in A,B should show the same distribution of values as in C,D. However, they are actually the opposite sign it seems (could it be that one is log₂ (A,B) and the other -log₂ (C,D)?). Please clarify and if possible, use the same values. Also, at the bottom of the legend it says that "14N FA is standardized to median of 12M group" - this does not seem to be the case for A to D and I.

8. Figure 3I: Is the data from both sexes or only one?

9. Figure 4B: It is probably my lack of knowledge, but what are the green spots on the right to the western blot (at least they are present in my version of the figure)?

10. Figure 4D and legend. Please define what NSAF stands for.

11. Figure 4G shows turnover values, which have been defined in the text and Figure 2A as " $14N / (14N + 15N)$ " - that means that at 18 months less 14N is left relative to total, so more protein has been turned over. However, the legend says that you see a "decrease in turnover". I do think it is exactly the opposite for the above reason, but could easily misunderstand. Please clarify.

12. Suppl. Figure 4: As before - if in B all the values are really standardized to the median of 12 months, why is then the median for 12 months well below a \log_2 value of 0? Again, I might misunderstand what the authors mean by "standardized to the median of 12 months", but in any case, a clarification would help.

13. I am missing a reference to Kluever et al., Science Advances, 2022 (PMID: 35594347). I think this paper is especially relevant in the context of this manuscript as it looks at protein turnover in young and old animals and especially in the brain. Therefore, the authors might consider referencing it in the manuscript.

Reviewer #3:

In this paper, the authors report a fine-grained study of protein turnover rates measured by metabolic labelling in vivo of three organs.

The study extends previous results from other groups that have performed similar studies on a smaller scale in the brain and points to a role of the proteasome in the regulation of protein turnover in the brain.

General point:

A clear hypothesis emerges from the data: reduced UPS activity after 18 months of age is responsible for the changes in the turnover of the catalytic core subunits of the proteasome observed at later ages. This hypothesis is testable by treating mice of 18 months of age with proteasome inhibitors (e.g. bortezomib) in order directly assess whether reduced proteasome activity is sufficient to mimic the aging-associated changes in turnover of the catalytic subunits. In my opinion, this experiment is a necessary functional validation to warrant publication in a prestigious journal such as Mol Syst Biol.

Specific points

1. Is there a correlation between protein abundance and protein turnover rates? In principle, it is expected that the most abundant proteins should have lower turnover rates.

2. It is known that myelin is subject to phagocytosis by microglia (Nat Neurosci . 2016 Aug;19(8):995-8. doi: 10.1038/nn.4325). Can this explain the changes in the turnover rates of myelin proteins?

3. The abundance of proteins in the insoluble fraction should be normalized with respect to the abundance of the same protein in the soluble fraction. In other words, it is expected that highly-abundant proteins are also abundant in the insoluble fraction.

4. Related to Fig. 3 C and D, the GO overrepresentation of the two "pink" clusters should be shown separately

5. By comparing Fig. 3 C and D, it appears that the "dynamic" clusters in the Fig. 3 D are comparable to the "static" clusters in Fig. 3 C. It is necessary to show confidence interval for the three clusters and to provide a statistical assessment of the their dynamic nature (e.g. via two-way ANOVA)

Responses to Reviewers' Comments

MSB-2023-11808

Reviewer #1:

Rao et al. conducted continuous in vivo metabolic labeling to investigate age-related changes in protein turnover in mice. They compared three tissues - two postmitotic and one mitotic. As expected, the mitotic tissue exhibited greater label incorporation than the postmitotic ones. However, only the cortex displayed age-dependent differences. Intrigued by the findings in the cortex, the researchers delved deeper and performed additional measurements, this time comparing age-related protein turnover changes in males and females. Surprisingly, while proteins from the same cellular compartments exhibited shifts in turnover due to aging, the shifts in female mice occurred at later time points compared to male mice. Part of the age-related shift in protein dynamics could be attributed to the formation of protein aggregates that evade degradation. Interestingly, not all compartments that displayed age-related turnover changes were represented in these insolublesome. Furthermore, the proteasome, a critical player in protein degradation, exhibited differential activity throughout aging. The regulatory proteasome subunits (19S) demonstrated a distinct activity pattern from the core proteasome (20S).

Overall, while this study is descriptive in nature, it presents a fascinating dataset that will undoubtedly interest various scientific communities, including those focused on aging, neurodegeneration, and protein turnover. The data quality appears to be high, and although additional experiments may not be necessary in principle, certain terms and statements in the writing require further clarification. Addressing these points would be beneficial before acceptance.

Response: We thank the Reviewer for carefully reading our manuscript and the positive assessment of our work. We appreciate your comments on the interesting yet descriptive nature of this manuscript. To make our manuscript even stronger we have performed an additional mechanistic experiment to functionally verify our result.

1. Figure 1B and C and also the corresponding text in the results section (2nd page of results, 2nd paragraph): The authors use the term "protein abundance" here. This is a bit misleading as they mean number of identified proteins, while nowadays protein abundance means more protein amount of the proteins. The wording should be adjusted accordingly. This is also important in light of Figure 1D where not the number of IDs are used to calculate FA, but the integrated MS signals, which is more in line with an abundance definition in proteomics.

Response: Thank you for this constructive comment. We agree that use of term ""protein abundance" is not suitable here. To address this concern, we have modified the figure 1B and C and the corresponding legend and text (please see line 127-130)

2. Figure 1B: Is the number of ID'd proteins in heart not significantly lower than in cortex or liver? Why was not the same cross-sample comparison performed as in 1C?

Response: We thank the reviewer for this suggestion to perform cross sample comparison in Figure 1B. This analysis has been performed and shows that the N15 protein IDs in heart tissue is significantly lower across aging compared to liver. This analysis has been added to Figure 1B.

3. In Figure 1 was the data generated from female and male animals or only from one sex? Please,

add the information to the legend. Same for Suppl. Figure 1.

Response: The data generated in Figure 1 is from female cohorts. This information has been added in the legend of Figure 1 and S1 and text (please see line 115).

4. Figure 2B and C: Relative ¹⁵N signal to WHAT? Is it to the median of 12 months as indicated at the bottom of the figure for ¹⁴N FA? If to the median of 12 months for each sex then why is the 12 months median for example at 2C not at a log₂ value of 0? This is throughout all the figures (see more below). Please define relative to WHAT as it is ambiguous sometimes. Moreover, in several figures it does not seem that it is true that it was standardized to the median of 12 months as then the median of 12 months (assuming the line in the boxplot is the median) should be at 1 in linear space or 0 in log₂ space. I might be mistaken what you mean by "standardized to the median of 12 months", but in any case, a clarification would help.

Response: We thank the Reviewer for the thoughtful comment and acknowledge that the word choices used to describe our data in the first submission had several limitations. The Reviewer is correct, our previous use of 'relative' is not accurate for Figure 2 B and C. In the revised manuscript, we removed 'relative' from the Y-axes in Figures 2B-C, the figure legend, and manuscript text. We can also see how our use of the term "standardization" caused unnecessary confusion. In order to address this limitation, we added an explicit description of the data presented in each figure legend.

We chose against normalizing directly to the 12 M time point for each respective protein. Had we gone that route, then we would be left with values of "1" for all proteins at 12 M, which would prevent proper statistical analyses. The boxplot line for all the plots is mean +/- SEM.

For Figure 2B-C, we extracted the TMT reporter ion intensities for each identified ¹⁵N protein and plotted them based on a log₂ scale. Next, we calculated the median value for the 12 M timepoint point and adjusted each protein based on this value.

For Figure 2D-E, we extracted the TMT reporter ion intensities for each protein identified in both the ¹⁴N and ¹⁵N channels and calculated the FA (i.e., (¹⁴N TMT intensity / (¹⁴N TMT intensity + ¹⁵N TMT intensity))). Next, we calculated the median FA value for the 12 M time point and adjusted each protein based on this value. This allowed for proper statistical analyses to be performed to compare all groups across aging.

The data presented in Figure 2F-G was prepared as in Figure 2D-E, but the FA value was plotted as scaled by row for heatmap representation.

5. Figure 2D, E: It says that "relative ¹⁴N protein abundance from both male and female cohorts NO significant differences..." I think the figure shows the opposite here - that there is a significant difference, doesn't it? Please correct.

Response: We apologize for this mistake; it was a typo and thank the Reviewer for catching this. This has been corrected in Figure 2 legend (please see line 713).

6. Figure 2G: K-means clustering. Honestly, I see here more than 2 clusters for the female data. Please explain the rationale, why only 2 clusters were chosen as parameter for the female mice?

Response: I can see why it may seem as if the female dataset contains more than 2 clusters, however, the male and female k-mean clustering analysis was performed with identical parameters. We have chosen 'k' (i.e., number of clusters) based on the optimal silhouette

score. Using the Orange bioinformatics pipeline, we ran 300 iterations to systematically determine how many clusters is ideal while also avoiding overfitting the data. For the female dataset, the optimal number of clusters calculated was 2.

7. Figures 3 A, B and C, D - based on the y-axis label the plots in A,B should show the same distribution of values as in C,D. However, they are actually the opposite sign it seems (could it be that one is $\log_2(A,B)$ and the other $-\log_2(C,D)$?). Please clarify and if possible, use the same values. Also, at the bottom of the legend it says that " ^{14}N FA is standardized to median of 12M group" - this does not seem to be the case for A to D and I.

Response: Thank you for this useful comment for improving the Figure representation. We have now presented the ^{14}N FA data on the scale of 0 to 1, as these values are calculated based on reconstructed MS1 chromatograms ($^{14}\text{N} / ^{14}\text{N} + ^{15}\text{N}$).

8. Figure 3I: Is the data from both sexes or only one?

Response: The protein ubiquitination data presented in Figure 3 (Figure 3G in the revised manuscript) is from male cohorts. This information has been added to the figure legend and text (please see line 255 and 748).

9. Figure 4B: It is probably my lack of knowledge, but what are the green spots on the right to the western blot (at least they are present in my version of the figure)?

Response: Sorry for the confusion, we have removed the cartoons of 20S and 19S proteasome complexes from the modified figures.

10. Figure 4D and legend. Please define what NSAF stands for.

Response: NSAF stands for Normalized Spectral Abundance Factor. This term was previously defined in PMC3599300. We have now defined NSAF in Figure 4 legend and modified the figured accordingly.

11. Figure 4G shows turnover values, which have been defined in the text and Figure 2A as " $^{14}\text{N} / (^{14}\text{N} + ^{15}\text{N})$ " - that means that at 18 months less ^{14}N is left relative to total, so more protein has been turned over. However, the legend says that you see a "decrease in turnover". I do think it is exactly the opposite for the above reason, but could easily misunderstand. Please clarify.

Response: We thank the reviewer for this helpful comment. We agree that the use of the term 'protein turnover' is confusing. To resolve this throughout the manuscript, we have modified the figures to say " ^{14}N fractional abundance (FA)", where $\text{FA} = (^{14}\text{N} / ^{14}\text{N} + ^{15}\text{N})$, instead of "protein turnover".

12. Suppl. Figure 4: As before - if in B all the values are really standardized to the median of 12 months, why is then the median for 12 months well below a \log_2 value of 0? Again, I might misunderstand what the authors mean by "standardized to the median of 12 months", but in any case, a clarification would help.

Response: We have removed this panel from Fig EV4.

13. I am missing a reference to Kluever et al., Science Advances, 2022 (PMID: 35594347). I think this paper is especially relevant in the context of this manuscript as it looks at protein turnover in young and old animals and especially in the brain. Therefore, the authors might consider referencing it in the

manuscript.

Response: We agree that the findings of *Kluever et al.* are very relevant to our manuscript and Eugenio Fornasiero is a friend. We apologize for this oversight and have add this citation. (please see lines 59, 315, 368, and 491)

Reviewer #3:

In this paper, the authors report a fine-grained study of protein turnover rates measured by metabolic labelling in vivo of three organs.

The study extends previous results from other groups that have performed similar studies on a smaller scale in the brain and points to a role of the proteasome in the regulation of protein turnover in the brain.

General point:

*A clear hypothesis emerges from the data: reduced UPS activity after 18 months of age is responsible for the changes in the turnover of the catalytic core subunits of the proteasome observed at later ages. This hypothesis is testable by treating mice of 18 months of age with proteasome inhibitors (e.g. bortezomib) in order directly assess whether reduced proteasome activity is sufficient to mimic the aging-associated changes in turnover of the catalytic subunits. In my opinion, this experiment is a necessary functional validation to warrant publication in a prestigious journal such as *Mol Syst Biol.**

Response: We thank the Review for this helpful improvement to our manuscript. We agree that showing that manipulating proteasome activity modulates proteasome subunit turnover will strengthen this work. We have performed another *in vivo* stable isotope labeling experiment in 18 M aged animals in combination with marizomib (a brain penetrant Psmb1/2/5 inhibitor). We have performed several follow up experiments and presented this new data in Figure 5/ Figure EV5.

Specific points

1. Is there a correlation between protein abundance and protein turnover rates? In principle, it is expected that the most abundant proteins should have lower turnover rates.

Response: We appreciate the Review's thoughtful comments on the relationship between protein abundance and protein turnover. We agree some highly abundant proteins are long-lived, however not every high-abundance protein is, by default, a long-lived protein (LLP), and not every low-abundance protein is short-lived (Figure 1, left).

Figure 1. Comparison of protein abundance based on ¹⁴N and ¹⁵N spectral counts.

Here we show the percent of old protein based on spectral counts (i.e. ^{14}N spectra # / (^{14}N + ^{15}N spectra #)*100) for the top 150 most abundant mitochondrial proteins (Bomba Warczak et al. PMID: 34259807), rank ordered according to number of total identified spectral counts ranging from 42 to 1064. As illustrated, for many of the highly abundant proteins, percent of ^{14}N -spectra counts are low and for some even zero, indicating that in spite of being abundant, these proteins are not LLPs. Similarly, the correlation between ^{14}N -spectral counts and total spectral abundance for the top 150 most abundant mitochondrial proteins is low, with an $R^2=0.1495$ (Figure 1, right).

For this work, we have below a similar example in which we have plotted the relationship between each protein's abundance and the same proteins turnover using MS1 based fractional abundance ($^{14}\text{N}/^{15}\text{N}+^{14}\text{N}$). The blue "x" symbols show rank ordered protein abundance (high to low) and the pink dots show their respective ^{14}N FA. Highly abundant proteins do not necessarily have lower turnover rates which is shown by the variable distribution of pink dots.

Figure 2. Comparison of protein abundance based on spectral counts compared to MS1-based fractional abundance.

2. *It is known that myelin is subject to phagocytosis by microglia (Nat Neurosci . 2016 Aug;19(8):995-8. doi: 10.1038/nn.4325). Can this explain the changes in the turnover rates of myelin proteins?*

Response: Thank you for this helpful suggestion. We have added discussion about myelin proteins turnover rates and cited the paper suggested by the Reviewer (please see line 403).

3. *The abundance of proteins in the insoluble fraction should be normalized with respect to the abundance of the same protein in the soluble fraction. In other words, it is expected that highly-abundant proteins are also abundant in the insoluble fraction.*

Response: The Reviewer raises an important point.

Please note, we never exclusively analyzed the "soluble fraction" of the proteome. For Figures 1 and 2, the tissues were thawed on ice and homogenized using a bead-based Precellys in 500 μL of homogenization buffer (4 mM HEPES, 0.32 M sucrose, 0.1 mM MgCl_2 with Halt protease inhibitor cocktail, and 1 mM PMSF) and methanol and chloroform precipitated. The pellets were solubilized in 8 M urea or 6M GuHCl, reduced, alkylated, and digested with trypsin or trypsin Lys-C. Thus, we never extracted the soluble fraction.

For the insolubleome analysis presented in Figure 3, cortical homogenates were sonicated and incubated with 1% SDS and the insoluble material was isolated by ultracentrifugation at 100,000 x g at 4 °C for 60 minutes. Our thinking was that maybe the differences in protein turnover identified across aging in Figure 2 are driven by misfolded proteins. By again using

the tissues from the continuously ¹⁵N labeled mice, we are able to calculate the ¹⁴N (old) protein level relative to the total insoluble pool to each respective protein. This provides a robust method for overcoming any variability in biochemical separation of insoluble fractions.

To further address the Reviewer's comment we have also extracted high and low abundance proteins in the insolubleome (Table 1), and presented their FA values from insolubleome and global analysis. These are examples of proteins who's higher FA value in the insolubleome is indicative of a deficit in turnover due to phase-separation/misfolding. We have provided the list of proteins that possess higher FA values in the insolubleome compared to global (or not identified in global) across times points in the Dataset EV3 and have provided examples below as well.

4.

12 M	insol	global	15 M	insol	global	18 M	insol	global	21 M	insol	global	24 M	insol	global
Eef1a1	0.017	#N/A	Oxct1	0.010	0.007	Gapdh	0.010	0.004	Gstp1	0.020	0.013	Acta1	0.026	0.023
Actg1	0.023	0.009	Actc1	0.024	0.021	Actc1	0.025	0.023	Actg1	0.026	0.021	Actg1	0.027	0.025
Actb	0.023	0.020	Acta1	0.028	0.019	Acta1	0.025	0.023	Actb	0.026	0.021	Actb	0.027	0.025
Actc1	0.024	0.019	Actg1	0.029	0.020	Actg1	0.025	0.022	Actc1	0.028	0.024	Atp1a1	0.036	0.033
Acta1	0.024	0.023	Actb	0.029	0.020	Actb	0.025	0.022	Acta1	0.028	0.024	Pkm	0.039	0.038
Ndufa13	0.034	0.018	Tpi1	0.033	0.015	Pgam1	0.041	0.003	Pgam1	0.041	0.015	Sncb	0.045	0.005
Cend1	0.034	0.033	Gapdh	0.033	0.031	Cox4i1	0.042	0.029	Sh3gl2	0.043	0.042	Ywhaz	0.048	0.005
Gapdh	0.039	0.032	Snap91	0.042	0.032	Hba	0.045	0.012	Syt1	0.044	0.026	Atp1a3	0.055	0.036
Snap91	0.046	0.005	Thy1	0.046	0.039	Cltc	0.045	0.034	Tkt	0.045	0.021	Cltc	0.055	0.044
Hnrnpu	0.050	0.037	Ywhaz	0.055	0.035	Thy1	0.051	0.041	Ndufa9	0.047	0.044	Ppia	0.056	0.006
Bin1	0.054	0.007	Ndufa5	0.060	0.050	Etfb	0.053	0.021	Camk2a	0.047	0.021	Mdh1	0.060	0.030
Slc1a2	0.054	0.052	Slc1a2	0.064	0.025	Mdh1	0.060	0.017	Atp6v1a	0.052	0.044	Thy1	0.063	0.051
Stxbp1	0.055	0.019	Fh	0.075	0.064	Hspa12a	0.061	0.050	Ywhaz	0.052	0.005	Hspe1	0.066	0.064
Ywhaz	0.060	0.050	Ppp3ca	0.082	0.080	Atp1a2	0.069	0.044	Cf1	0.055	0.013	Stxbp1	0.069	0.056
Ctnn	0.062	0.037	Ckmt1	0.084	0.049	Slc1a2	0.069	0.021	Slc1a2	0.064	0.026	Atp6v0a1	0.072	0.024

Table 1. Comparison of FA values high and low abundant proteins identified in the insolubleome and the corresponding global proteome FA.

Response: Thank you for this helpful suggestion. In our revised manuscript, we now have presented the two pink clusters separately.

5. By comparing Fig. 3 C and D, it appears that the "dynamic" clusters in the Fig. 3 D are comparable to the "static" clusters in Fig. 3 C. It is necessary to show confidence interval for the three clusters and to provide a statistical assessment of the their dynamic nature (e.g. via two-way ANOVA)

Response: We thank the Reviewer for this critical suggestion. We agree that this is necessary to show the significant differences between the three clusters. Accordingly, we have performed the two-way ANOVA and have added it to the Figure 3.

8th Dec 2023

Manuscript Number: MSB-2023-11808R

Title: Derailed protein turnover in the aging mammalian brain.

Dear Jeff,

Thank you for sending us your revised manuscript. We have now heard back from the two reviewers who were asked to evaluate your revised study. As you will see below, the reviewers are satisfied with the performed revisions and support publication. As such, I am glad to inform you that we can soon accept your manuscript for publication, pending some minor revisions listed below, all related to editorial issues.

- Our Data Editors noted that the following needs to be corrected/added in the Figure Legends (main + EV):

-- Please note that the box plots need to be defined in terms of minima, maxima, centre, bounds of box and whiskers, and percentile in the legends of figures 2b-c; 4d; 5e

-- Please note that information related to n is missing in the legends of figures 5b; EV5f.

- The funding information provided in the manuscript text need to match the information entered in the online submission system. R21 AG080705 is currently missing from the submission system.

- Please remove the 'Authors Contributions' from the manuscript. The 'Author Contributions' section is replaced by the CRediT contributor roles taxonomy to specify the contributions of each author in the journal submission system. Please use the free text box in the 'author information' section of the online submission system to provide more detailed descriptions if needed (e.g., 'X provided intracellular Ca⁺⁺ measurements in fig Y').

- Please complete the Author Checklist.

- Source Data: Source Data: Please provide one file (or .zip folder) per main figure and a single .zip folder for all EV and/or Appendix figure Source Data.

- The legends for all figures (main and EV) should be placed after the References.

- The synopsis image does not display well at the final size required. Please resupply the image as a jpg or png at the required final size (it needs to be exactly 550 px wide, and the height ideally < 500 px), ensuring that all labels are legible.

Please resubmit your revised manuscript ****within one month**** and ideally as soon as possible. If we do not receive the revised manuscript within this time period, the file might be closed and any subsequent resubmission would be treated as a new manuscript. Please use the Manuscript Number (above) in all correspondence.

When you resubmit your manuscript, please download our CHECKLIST (<https://bit.ly/EMBOPressAuthorChecklist>) and include the completed form in your submission. *Please note* that the Author Checklist will be published alongside the paper as part of the transparent process (<https://www.embopress.org/page/journal/17444292/authorguide#transparentprocess>)

Click on the link below to submit your revised paper.

Kind regards,

Maria

Maria Polychronidou, PhD
Senior Editor
Molecular Systems Biology

If you do choose to resubmit, please click on the link below to submit the revision online before 7th Jan 2024.

IMPORTANT:

Please note that corresponding authors are required to supply an ORCID ID for their name upon submission of a revised manuscript (EMBO Press signed a joint statement to encourage ORCID adoption).

(<https://www.embopress.org/page/journal/17444292/authorguide#editorialprocess>)

Currently, our records indicate that the ORCID for your account is 0000-0002-8173-5580.

Link Not Available

*** PLEASE NOTE *** As part of the EMBO Press transparent editorial process initiative (see our Editorial at <https://dx.doi.org/10.1038/msb.2010.72> , Molecular Systems Biology will publish online a Review Process File to accompany accepted manuscripts. When preparing your letter of response, please be aware that in the event of acceptance, your cover letter/point-by-point document will be included as part of this File, which will be available to the scientific community. More information about this initiative is available in our Instructions to Authors. If you have any questions about this initiative, please contact the editorial office (msb@embo.org).

Reviewer #1:

The authors have addressed all my concerns and I fully support publication of this manuscript.

Reviewer #3:

My points have been fully addressed and I have no further comments.

All editorial and formatting issues were resolved by the authors.

13th Dec 2023

Manuscript number: MSB-2023-11808RR

Title: Derailed protein turnover in the aging mammalian brain.

Dear Jeff,

Thank you again for sending us your revised manuscript. We are now satisfied with the modifications made and I am pleased to inform you that your paper has been accepted for publication.

Kind regards,

Maria

Maria Polychronidou, PhD
Senior Editor
Molecular Systems Biology
